# A fungal core effector exploits the OsPUX8B.2–OsCDC48-6 module to suppress plant immunity

Xuetao Shi[1,2], Xin Xie[1], Yuanwen Guo[1], Junqi Zhang[1], Ziwen Gong[1,2], Kai Zhang[1], Jie Mei[1,2], Xinyao Xia[1], Haoxue Xia[1], Na Ning[1], Yutao Xiao [2], Qing Yang [1], Guo-Liang Wang [3] & Wende Liu [1] ✉

Proteins containing a ubiquitin regulatory X (UBX) domain are cofactors of Cell Division Cycle 48 (CDC48) and function in protein quality control. However, whether and how UBX-containing proteins participate in host–microbe interactions remain unclear. Here we show that MoNLE1, an effector from the fungal pathogen *Magnaporthe oryzae*, is a core virulence factor that suppresses rice immunity by specifically interfering with OsPUX8B.2. The UBX domain of OsPUX8B.2 is required for its binding to OsATG8 and OsCDC48-6 and controls its 26 S proteasome–dependent stability. OsPUX8B.2 and OsCDC48-6 positively regulate plant immunity against blast fungus, while the high-temperature tolerance heat-shock protein OsBHT, a putative cytoplasmic substrate of OsPUX8B.2–OsCDC48-6, negatively regulates defense against blast infection. MoNLE1 promotes the nuclear migration and degradation of OsPUX8B.2 and disturbs its association with OsBHT. Given the high conservation of MoNLE1 among fungal isolates, plants with broad and durable blast resistance might be generated by engineering intracellular proteins resistant to MoNLE1.

Ubiquitin regulatory X (UBX) proteins, which contain a UBX domain of approximately 80 amino acids mimicking a ubiquitin fold, constitute the largest group of cofactors of Cell Division Cycle 48 (CDC48, also called p97 or valosin-containing protein [VCP] in metazoans)[1]. Hexamers of CDC48, an AAA+ ATPase family member, function as "segregases" or "unfoldases" that extract proteins from organelle membranes or protein complexes; the extracted proteins are subsequently degraded by the 26S proteasome[2–4]. CDC48 cofactors regulate the processing of substrates or serve as adapters that determine substrate specificity. In yeast (*Saccharomyces cerevisiae*) and mammalian cells, UBX proteins are involved in degradation pathways, such as endoplasmic reticulum (ER)–associated degradation[5], mitochondrion-associated degradation[6], lipid droplet homeostasis[7], and stress granule

disassembly[8]. Emerging evidence suggests that UBX family members in humans bind to specific proteins involved in cancer cell proliferation, making them potential therapeutic targets for inhibiting tumor growth[9]. For example, UBXN2A directly inhibits the oncoprotein mortalin-2 and induces cell apoptosis to suppress tumor growth[10–13]; similarly, the UBX protein p47 antagonizes the nuclear factor-κB (NF-κB) pathway that is activated in tumor cells by degrading the NF-κB essential modulator (NEMO)[14].

Plant UBX-containing proteins (PUXs) have primarily been studied in the model plant Arabidopsis (*Arabidopsis thaliana*). Arabidopsis has 16 PUX family members (PUX1–16), 10 of which have been functionally characterized. PUX1 regulates the activity of CDC48 by affecting the oligomeric status of CDC48[15–17]. PUX2 is the only family member

[1]State Key Laboratory for Biology of Plant Diseases and Insect Pests, Institute of Plant Protection, Chinese Academy of Agricultural Sciences, Beijing 100193, China. [2]Shenzhen Branch, Guangdong Laboratory of Lingnan Modern Agriculture, Key Laboratory of Gene Editing Technologies (Hainan), Ministry of Agriculture and Rural Affairs, Agricultural Genomics Institute at Shenzhen, Chinese Academy of Agricultural Sciences, Shenzhen 518000, China. [3]Department of Plant Pathology, The Ohio State University, Columbus, OH 43210, USA. ✉e-mail: liuwende@caas.cn

reported to be involved in pathogen defense, likely via its function in host cell ploidy[18,19]. PUX10 mediates the turnover of lipid droplet–related proteins via CDC48[20,21]. PUX7, PUX8, PUX9, and PUX13 are autophagy adapters for the clearance of nonfunctional CDC48 proteins[22], and PUX3, PUX4, and PUX5 associate with CDC48 but negatively regulate the stability of the inner nuclear membrane protein SAD1/UNC-84 DOMAIN PROTEIN 1 (SUN1)[23]. However, little is known about the complement and functions of PUX proteins in other plants, especially in crops. Moreover, although compelling evidence suggests that CDC48s are essential regulators of plant immunity[24], little is known about whether and how the cofactors of CDC48, such as PUX proteins, participate in immune responses.

The pathogen *Magnaporthe oryzae* (synonym *Pyricularia oryzae*) causes devastating blast disease in monocot crops including rice (*Oryza sativa*), wheat (*Triticum aestivum*), finger millet (*Eleusine coracana*), and foxtail millet (*Setaria italica*)[25]. Successful invasion by *M. oryzae* requires a repertoire of effectors that are secreted into the cytoplasmic or apoplastic spaces of host cells and suppress host immune responses. Several of these effectors influence host protein homeostasis. For instance, the *M. oryzae* avirulence factor AvrPiz-t targets the rice E3 ligases AvrPiz-t INTERACTING PROTEIN 6 (APIP6) and APIP10, thereby suppressing their E3 ligase activity and promoting their degradation through the 26S proteasome[26,27]. AvrPi9 was recently reported to stabilize the ubiquitin-like domain-containing protein AVRPI9-INTERACTING PROTEIN 1 (ANIP1) and facilitate the degradation of the ANIP1 substrate OsWRKY62[28]. However, the mechanisms of most *M. oryzae* effectors remain elusive due to their small size and lack of conserved domains.

Here, we identified a core effector of *M. oryzae* named NUCLEUS-LOCALIZED EFFECTOR 1 (MoNLE1), which localizes to the nucleus of the plant host and functions as a virulence effector that suppresses basal immunity in rice. MoNLE1 specifically interacts with OsPUX8B.2, a positive regulator of the defense response against *M. oryzae*. OsPUX8B.2 acts as a cofactor of OsCDC48-6 that mediates the turnover of the cytoplasmic heat-shock protein BOOTING STAGE HIGH-TEMPERATURE TOLERANCE (OsBHT), a negative regulator of plant immunity against *M. oryzae*. We determined that MoNLE1 not only promotes the nuclear migration and degradation of OsPUX8B.2, it also disrupts the association between OsPUX8B.2 and OsBHT, shedding light on the virulence mechanism of this *M. oryzae* effector.

## Results

### MoNLE1 is a putative core nuclear virulence effector

We previously identified 851 fungal genes expressed *in planta* during fungal infection that encode putative secreted proteins during the interaction between *M. oryzae* and rice[29]. As effectors that relocate to specific organelles in the host might have distinct functions, we focused on effectors that target the host nucleus in this study. We used NLS mapper (https://nls-mapper.iab.keio.ac.jp/cgi-bin/NLS_Mapper_form.cgi) to predict classical nuclear localization signals (cNLSs) among the secreted proteins encoded by the 851 genes expressed *in planta* and identified 49 putative effectors harboring high-confidence cNLSs (cutoff score ≥6; Supplementary Data 1). Only three proteins (*M. oryzae* Host Transcription Reprogramming 1 [MoHTR1] and two proteins of unknown function) overlapped with previously identified effectors with transcriptional activity (Supplementary Fig. 1a and Supplementary Data 1)[30], suggesting that nuclear effectors have diverse functions. In parallel, we performed a conservation analysis of effectors using 187 existing *M. oryzae* genomes (Supplementary Data 2), which returned six highly conserved putative effectors in *Oryza* pathotype strains (Supplementary Fig. 1b). Among the six effectors, only one protein (MGG_04776), which we named NUCLEUS-LOCALIZED EFFECTOR 1 (MoNLE1), was in the list of 49 putative nuclear effectors (Fig. 1a and Supplementary Fig. 1a, b).

To evaluate the variation in *MoNLE1* among *M. oryzae* strains, we examined its genome sequence in 187 strains from 13 different pathotypes (Supplementary Data 2). We reconstructed a gene-based phylogenetic tree to examine the relationships among *M. oryzae* isolates with respect to *MoNLE1* (Fig. 1b). The tree revealed distinct clusters of *MoNLE1* associated with specific hosts, most notably *Oryza*. *MoNLE1* was classified into three clades (clades 1–3). Most isolates from the rice lineage clustered into clade 1, in which *MoNLE1* appeared to be highly conserved, sharing nearly 100% identity with the other members. Clade 2 comprised most of the non-rice isolates, such as wheat, millet, and other grass isolates; members of clade 2 could be distinguished from those of clade 1 by the presence of a 204-bp insertion within *MoNLE1* (with similarity scores of 88–92%). Clade 3 included only two isolates from grasses that were genetically distinct from those of the other two clades, with a 204-bp deletion compared with strains from clade 2 and an additional 18-bp deletion when compared with clade 1 strains (Fig. 1b). A synteny plot drawn between rice isolate 70-15 and its homologs in *Triticum* and *Eleusine* pathotype isolates illustrated a series of complex structural rearrangements spanning ~400 kb in the vicinity of the *MoNLE1* locus (Supplementary Fig. 1c). These structural variations are characterized by extensive non-colinear alignments and recombination events, which might have acted as a driving force for the rapid evolution of neighboring genomic regions (Supplementary Fig. 1c). These results indicate that *MoNLE1* is highly conserved in *M. oryzae* isolates from different plants of the rice lineage.

*MoNLE1* was expressed at low levels in plate-cultivated mycelium and conidia but was induced following the infection of rice plants; it reached its highest expression level at 24 h post inoculation (hpi), followed by a decrease (Fig. 1c), indicating that MoNLE1 mainly functions at the early infection stage. We investigated whether MoNLE1 is essential for *M. oryzae* infection by generating a *MoNLE1*-knockout (*ko*) mutant in a *M. oryzae* strain (GUY11 background) and testing its pathogenicity on rice (Supplementary Fig. 1d, e). The *monle1 ko* mutant showed severe defects in inducing lesions on rice leaves, which were rescued in complementation strains harboring an intact copy of the *MoNLE1* gene in the *monle1* mutant (Fig. 1d and Supplementary Fig. 1d, e). Moreover, the blast strain (GUY11 background) over-expressing *MoNLE1-GFP* (encoding full-length MoNLE1 fused to a C-terminal green fluorescent protein) driven by the *RIBOSOMAL PROTEIN 27* promoter (*RP27p*) showed enhanced pathogenicity on rice plants (Supplementary Fig. 1f, g). These results indicate that *MoNLE1* plays a key role in full pathogen virulence.

### MoNLE1 targets the host nucleus and sabotages rice immunity by suppressing PAMP-triggered immunity

*MoNLE1* encodes a 172–amino acid (aa) protein with a putative 21-aa signal peptide (SP) for secretion at its N terminus and a cNLS at its C terminus (Fig. 1a). Using a yeast trap assay[31], we determined that the SP of MoNLE1 is functional and capable of mediating the secretion of this effector (Supplementary Fig. 1h, i). To explore the movement of MoNLE1 during pathogen infection, we inoculated rice sheaths using the *M. oryzae* strain overexpressing *MoNLE1-GFP*. We observed MoNLE1-GFP in structures mimicking biotrophic interfacial complexes (BICs), a hallmark of secretion for cytoplasmic effectors[32], at 30 hpi (Fig. 1e). We also observed a clear accumulation of MoNLE1-GFP in the host nucleus of rice sheath cells at 42 hpi (Fig. 1e). The transfection of rice protoplasts with constructs expressing *MoNLE1-GFP* (unless otherwise indicated, all following experiments were performed with MoNLE1 lacking the SP) confirmed that MoNLE1 is indeed a host nucleus–localized effector protein, as evidenced by fluorescence microscopy (Fig. 1f).

To examine the function of MoNLE1 during *M. oryzae* infection, we generated transgenic rice plants heterologous expressing *MoNLE1-HA* (encoding a fusion between MoNLE1 and a C-terminal 4 × HA tag, driven by the maize *Ubiquitin* promoter [*Ubip*]) in the rice

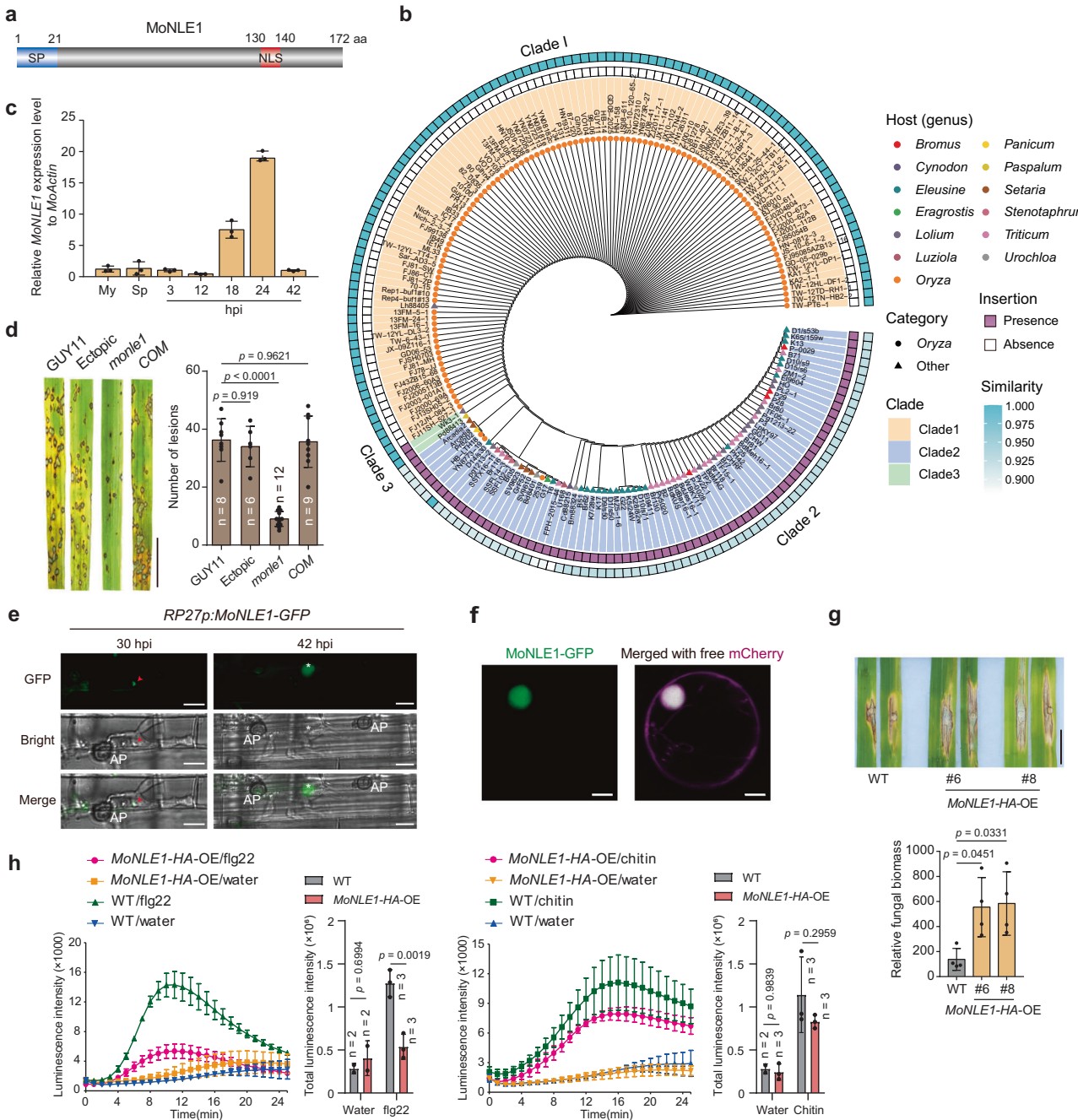

**Fig. 1 | MoNLE1 is a host nucleus–localized core virulence effector. a** Domain organization of MoNLE1. SP, signal peptide; NLS, nuclear localization signal; aa, amino acids. **b** Phylogenetic analysis of *MoNLE1* from rice, wheat, and grass isolates of *Magnaporthe oryzae*. **c** Relative *MoNLE1* expression level in mycelia (My), spores (Sp), and plants at the indicated time points. Data are means ± s.d. from three biological replicates. hpi, hours post inoculation. **d** Pathogenicity test of *monle1* and complementation (COM) strains. GUY11 (wild type, WT) and a strain harboring an ectopic insertion of the *HPT* disruption cassette (Ectopic) were used as controls. Data are means ± s.d. from the indicated number of biological replicates. Scale bar, 1 cm. **e** Subcellular localization of MoNLE1-GFP in the cells of rice sheaths during *M. oryzae* infection. Red arrowheads indicate the biotrophic interfacial complex (BIC); asterisks indicate host cell nuclei. AP, appressorium. Scale bars, 5 μm. **f** Subcellular

localization of MoNLE1-GFP in rice protoplasts. Free mCherry was used to label the nucleus and cytoplasm. **g** Heterologous expression of *MoNLE1-HA* (*Ubip:MoNLE1-HA*) decreases immunity in rice. The punch inoculation method was used to determine the disease phenotype; images were taken 9 days post inoculation (dpi). Relative fungal biomass was used to quantify disease symptoms. Data are means ± s.d. from four biological replicates. Scale bar, 1 cm. **h** Time-course accumulation of reactive oxygen species (ROS) and cumulative levels in *Ubip:MoNLE1-HA* (line #6) and wild-type (NPB) rice plants upon treatment with flg22 or chitin. Data are means ± s.e.m. from the indicated number of biological replicates. Data were analyzed by one-way (**d**, **g**) or two-way (**h**) analysis of variance (ANOVA) followed by Tukey's test, and the adjusted *p* values were shown in figures. Data in **e**, **f** are representatives of three independent experiments with similar results.

cultivar Nipponbare (NPB) to determine whether MoNLE1 affects plant immunity (Supplementary Fig. 1j). We inoculated the transgenic lines with *M. oryzae* strains RB22 and GUY11 using the punch inoculation method[33]. The *Ubip:MoNLE1-HA* plants showed increased susceptibility to rice blast infection (Fig. 1g and

Supplementary Fig. 1k), indicating that MoNLE1 facilitates *M. oryzae* infection in the host. Consistent with this result, the production of reactive oxygen species (ROS) induced by pathogen-associated molecular patterns (PAMPs) such as chitin and flg22 was also impaired in *Ubip:MoNLE1-HA* rice plants (Fig. 1h).

To investigate whether the NLS is required for the virulence function of MoNLE1, we generated a MoNLE1 variant, called MoNLE1[NES], by replacing the existing NLS with a nuclear export signal (NES). This largely excluded MoNLE1 from the nucleus, with this variant accumulating in the cytoplasm instead (Supplementary Fig. 2a). We generated *Ubip:MoNLE1[NES]-HA* transgenic plants and examined their phenotype upon infection with blast isolate RB22 (Supplementary Fig. 2b). The enhanced plant susceptibility seen in *Ubip:MoNLE1-HA* plants was largely compromised in *Ubip:MoNLE1[NES]-HA* lines, indicating that the NLS is essential for MoNLE1 function (Supplementary Fig. 2c). Taken together, these results indicate that MoNLE1 is a virulence effector of *M. oryzae* that interferes with host PAMP-triggered immunity, which is dependent on its NLS.

## MoNLE1 physically associates with rice OsPUX8B.2

To explore the virulence mechanism of MoNLE1 in the host, we looked for rice interactors of MoNLE1 by performing a yeast two-hybrid (Y2H) screen against a cDNA library generated using total RNA from *M. oryzae*–inoculated rice seedlings. We identified one MoNLE1 interactor, a PUX protein encoded by the locus LOC_Os09g35710. The rice genome encodes 12 rice PUX family members, among which OsPUX8A (LOC_Os08g43330) and OsPUX8B (LOC_Os09g35710) are homologous to Arabidopsis PUX8 (Supplementary Fig. 3a). The PUX protein identified by Y2H screening is encoded by one of the three possible *OsPUX8B* splice variants and was named OsPUXB.2 in this study. Compared with other *OsPUX* genes, *OsPUX8B* showed moderate expression levels in all rice tissues (Supplementary Fig. 3b). Homologs of OsPUX8B are widely distributed and cluster in different members of the green lineage, including monocots, eudicots, gymnosperms, and algae. Notably, nearly half of the homologous genes in monocot and eudicot plants are predicted to produce two to three transcripts (Supplementary Fig. 3c and Supplementary Data 3).

OsPUX8B.2 contains a ubiquitin-associated (UBA) domain at its N terminus, a ubiquitin-interacting motif (UIM) in its middle region, and a UBX domain at its C terminus (Fig. 2a and Supplementary Fig. 3d). OsPUX8B.2 differs from OsPUX8B.1 in its UBX domain, with minor changes at the end of the C terminus, and OsPUX8B.3 shares the same sequence as OsPUX8B.1 but lacks the UBA domain (Fig. 2b and Supplementary Fig. 3d). We also noticed differences in the UBX domain between the proteins encoded by splicing variants in *OsPUX8B* homologs of other plant species, such as maize (*Zea mays*), wheat, and Arabidopsis (Supplementary Fig. 3e), pointing to the possible functional differentiation of these proteins.

Y2H experiments showed that MoNLE1 interacts with the C terminus (aa 268–543) of OsPUX8B.2. The UBX domain was necessary for this interaction; however, the UBX domain of OsPUX8B.2 or OsPUX8B.1 alone was not sufficient for these proteins to interact (Fig. 2c and Supplementary Fig. 4a), indicating that regions outside the UBX domain are also needed for this interaction. Notably, the NLS of MoNLE1 was not required for the interaction, as deleting the NLS of MoNLE1 or replacing it with an NES did not prevent interaction (Fig. 2d and Supplementary Fig. 4b). MoNLE1 specifically interacted with OsPUX8B.2 but not with the other splicing variant products or with other homologs, as confirmed by Y2H and co-immunoprecipitation (Co-IP) assays (Fig. 2e, f and Supplementary Fig. 4c). Subcellular localization analysis in rice protoplasts showed that OsPUX8B.2-GFP is distributed in the nucleus and cytoplasm; the nuclear localization of OsPUX8B.2-GFP increased when the encoding construct was co-expressed with *MoNLE1-mCherry*, indicating that MoNLE1 promotes the migration of OsPUX8B.2 into the nucleus (Fig. 2g). To explore this result, we fractionated total proteins extracted from protoplasts co-transfected with *Ubipro:4 × HA-OsPUX8B.2* and *35S:GFP*, *35S:MoNLE1-GFP*, or *35S:MoNLE1[NES]-GFP* into total and nuclear proteins. The abundance of 4 × HA-OsPUX8B.2 in the nucleus was greater in the presence of MoNLE1-GFP than with

GFP, while MoNLE1[NES]-GFP resulted in lower OsPUXB.2 levels in nuclei, indicating that MoNLE1 indeed promotes the nuclear accumulation of OsPUX8B.2 through its NLS (Fig. 2h). Notably, similar to OsPUX8B.2, the subcellular localization of OsPUX8B.1 and OsPUX8B.3 was also affected by MoNLE1 (Supplementary Fig. 5a), suggesting that MoNLE1 may also disturb their localizations though common interactors.

In addition, MoNLE1 homologs in blast strains collected from wheat or grasses still interacted with OsPUX8B.2 (Fig. 2i and Supplementary Fig. 4d) despite the presence of an insertion in the encoding genes (Fig. 1c). Intriguingly, all MoNLE1 homologs examined also interacted with TaPUX8.2 encoded by one of the wheat splice variants lacking the UBX domain (Fig. 2j and Supplementary Figs. 3c and 4e), indicating that targeting OsPUX8B.2 homologs is a conserved function of MoNLE1. However, we only detected the interaction between MoNLE1 homologs and TaPUX8.2 under less stringent conditions in the Y2H assay (Fig. 2j), suggesting that these interactions might be relatively weak.

## OsPUX8B.2 is a positive defense regulator against *M. oryzae*

Under normal growth conditions, *OsPUX8B.2* exhibited the lowest expression level among the three transcript isoforms in all tissues examined: roots, shoots, and leaves (Supplementary Fig. 5b, c). Upon *M. oryzae* infection, the total *OsPUX8B* transcript level and the levels of all three individual *OsPUX8B* transcripts increased to a comparable extent during the early and later stages of infection (Supplementary Fig. 5d), suggesting that *M. oryzae* infection induces changes in overall *OsPUX8B* transcription rather than alternative splicing.

To determine whether OsPUX8B is involved in blast disease resistance, we generated *Ospux8b-ko* mutants using clustered regularly interspaced short palindromic repeats (CRISPR)/CRISPR-associated nuclease 9 (Cas9)-mediated gene editing, as well as plants overexpressing (OE) *OsPUX8B.1-HA*, *OsPUX8B.2-HA*, or *OsPUX8B.3-HA* driven by the maize *Ubipro* in the NPB background (Supplementary Fig. 5e–g). We challenged T₁ plants with the compatible *M. oryzae* isolate RB22 and did not observe significant differences in disease symptoms between *Ospux8b* mutants and wild-type plants (Fig. 2k). However, compared with the wild type, *OsPUX8B.2-HA*-OE plants displayed enhanced disease resistance, whereas *OsPUX8B.1-HA*-OE plants were more susceptible to the pathogen (Fig. 2k) and *OsPUX8B.3-HA*-OE plants did not show any significant difference (Supplementary Fig. 5h). These results suggest that OsPUX8B.1 and OsPUX8B.2 play opposite roles in immune responses and might compete for the same substrate(s). Considering the opposite phenotypes of *OsPUX8B.2-HA*-OE and *Ubip:MoNLE1-HA* plants against blast pathogen, this finding suggests that MoNLE1 might promote pathogen virulence by suppressing OsPUX8B.2 function.

## OsPUX8B.2 interacts with OsATG8 and OsCDC48-6 via its UBX domain

Arabidopsis PUX8 and its homologs PUX7, PUX9, and PUX13 function redundantly as adapters between AUTOPHAGY-RELATED 8 (ATG8) and nonfunctional CDC48 proteins[22]. ATG8 binds to the UIM of these PUXs via its UIM-docking site (UDS) and mediates the degradation of nonfunctional CDC48 via the autophagy pathway[22]. We thus examined whether OsPUX8B.2 interacts with rice ATG8 or CDC48 using Y2H experiments. OsPUX8B.2 interacted with all three rice ATG8 proteins tested (OsATG8a, OsATG8b, and OsATG8c) and one rice CDC48 (OsCDC48-6) among the six successfully cloned rice *CDC48* members (eight in total) (Fig. 3a, b and Supplementary Figs. 4f, g and 6a). The UBX domain but not the UIM of OsPUX8-2.2 was necessary for its interaction with the UDS of OsATG8a or with OsCDC48-6, as evidenced by Y2H, Co-IP, and split-luciferase complementation (SLC) assays (Fig. 3c–h and Supplementary Fig. 4h, i). Notably, similar to MoNLE1,

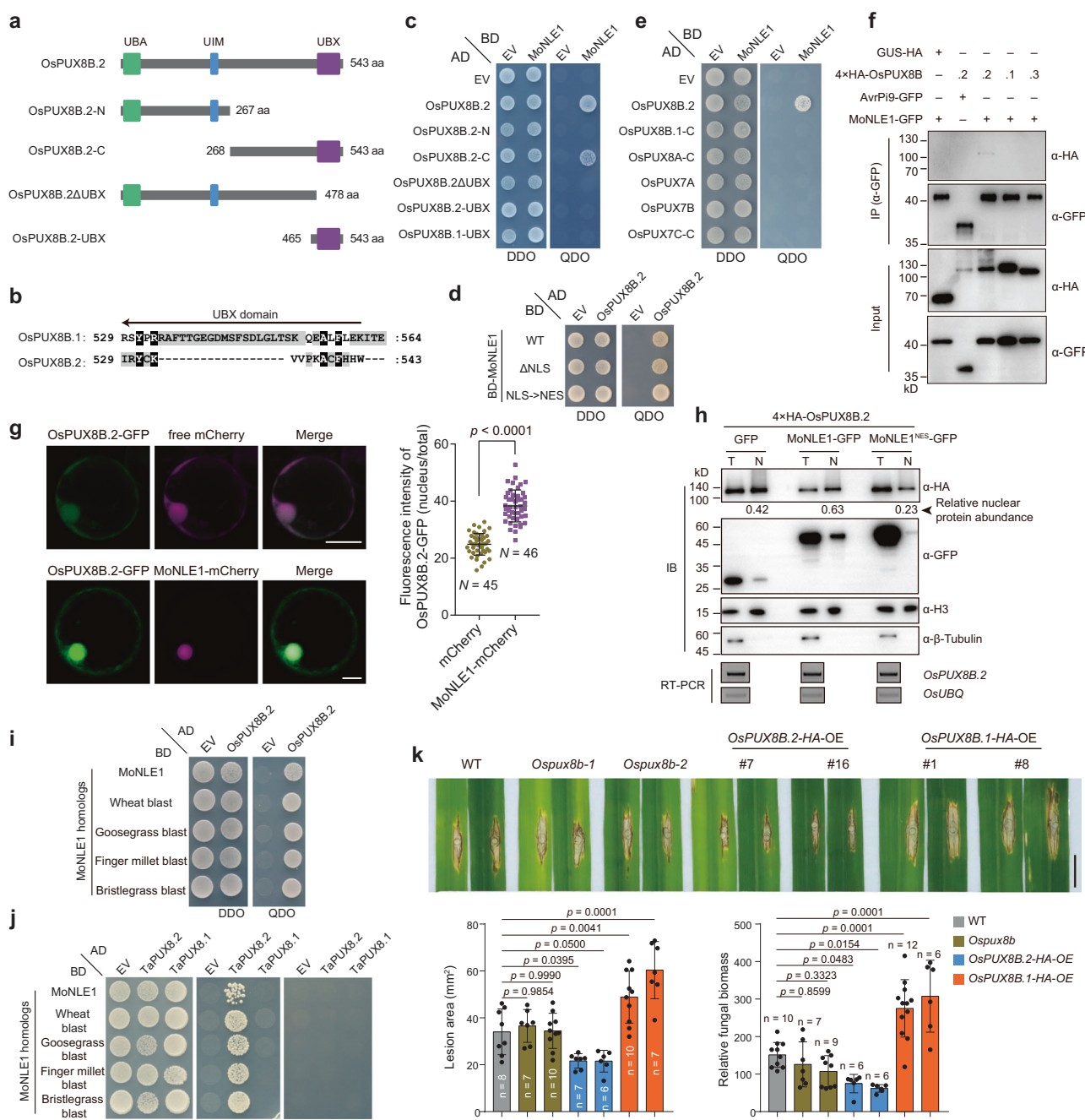

**Fig. 2 | MoNLE1 specifically targets rice OsPUX8B.2. a** Diagram of the OsPUX8B.2 fragments used in the yeast two-hybrid (Y2H) assay. UBA, ubiquitin-associated domain; UIM, ubiquitin interaction motif; UBX, ubiquitin regulatory X domain. **b** Sequence alignment of the UBX domain of OsPUX8B.1 and OsPUX8B.2. **c** Y2H assay to identify the interacting regions of MoNLE1 and OsPUX8B.2. DDO, double dropout medium (synthetic defined [SD]/−Leu/−Trp medium); QDO, quadruple dropout medium (SD/−Ade/−His/−Leu/−Trp medium). **d** Y2H assay examining the interaction of OsPUX8B.2 and MoNLE1, a variant lacking the nuclear localization signal (NLS), or a variant carrying a nuclear export signal (NES) in place of the NLS. **e** Y2H assay of the interaction specificity between MoNLE1 and OsPUX8B.2. Due to the apparent death of cells harboring the full-length constructs, the C-terminal regions of OsPUX8B.1 (aa 268–564), OsPUX8A (aa 289–582), and OsPUX7C (aa 116–233) were used in the Y2H assay. **f** Co-immunoprecipitation (Co-IP) assay to detect the interaction of MoNLE1 with different OsPUX8B proteins from different mRNA splicing variants. The effector protein AvrPi9 was used as a negative control. **g** Colocalization of OsPUX8B.2-GFP with free mCherry and MoNLE1-mCherry (E1-mCherry) in rice protoplasts. Ratio (nucleus/total) of OsPUX8B.2-GFP fluorescence intensities when its encoding construct was co-transfected with that encoding mCherry or MoNLE1-mCherry is shown in the right dot plot. Data are means ± s.d. from the indicated number of biological replicates. The $p$ value was determined by two-tailed unpaired Student's $t$ test. **h** Nuclear fractionation of proteins extracted from rice protoplasts producing the indicated proteins. Proteins were detected by immunoblot analysis (IB); RT-PCR analysis was used to assess *OsPUX8B.2* transcript levels. The relative nuclear protein abundance of 4×HA-OsPUX8B.2 was obtained by comparing the relative nuclear (N) protein level (4×HA-OsPUX8B.2$^N$ normalized to histone H3) to the relative total (T) protein level (4×HA-OsPUX8B.2$^T$ normalized to β-Tubulin) with ImageJ software. **i**, **j** Y2H assays showing the interaction between MoNLE1 homologs and OsPUX8B.2 (**h**) or TaPUX8.2 (**i**). TDO, triple dropout medium (SD/−His/−Leu/−Trp medium). **k** Phenotypes of *Ospux8b*, *OsPUX8B.2*-OE, and *OsPUX8B.1*-OE plants infected with blast fungus (strain RB22). Data were collected at 10 dpi (upper panel). Lesion area and relative fungal biomass (lower panel) were used to quantify disease symptoms. Data are means ± s.d. from the indicated number of biological replicates. Statistical analysis was performed by one-way ANOVA followed by Tukey's test, and the adjusted $p$ values were shown. Scale bar, 1 cm. Data in **f**, **h** are representatives of three independent experiments. Note that the protoplasts used in **f**–**h** were pretreated with MG132 for 10 h before collection.

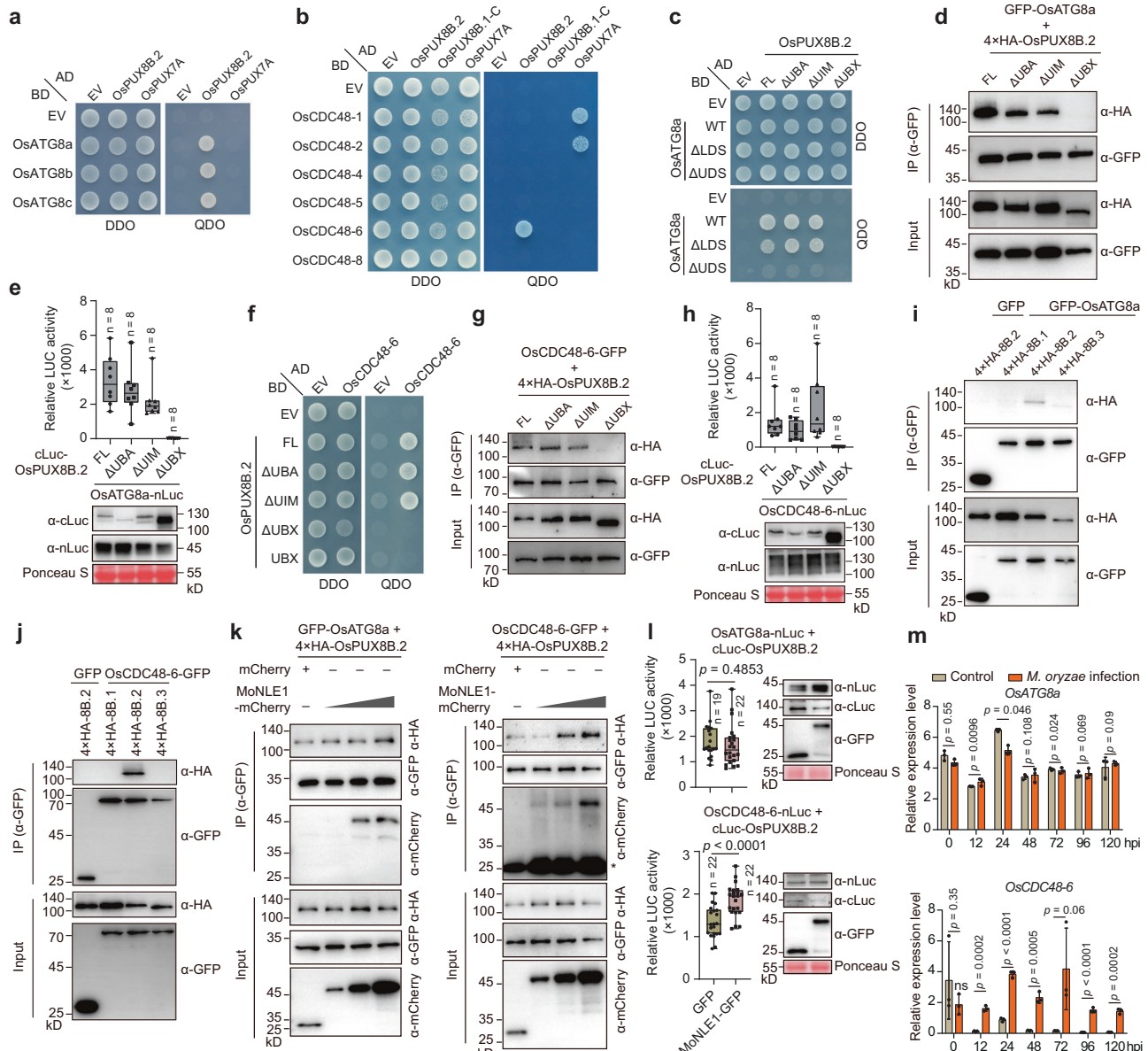

**Fig. 3 | OsPUX8B.2 associates with OsATG8a and OsCDC48-6 via its UBX domain. a, b** Y2H assays determining the interactions between OsPUX8B.2 and rice ATG8 (**a**) or CDC48 (**b**). **c**–**e** Y2H (**c**), Co-IP (**d**), and split-luciferase complementation (SLC) (**e**) assays determining the binding interface between OsATG8a and OsPUX8B.2. **f**–**h** Y2H (**e**), Co-IP (**f**), and SLC (**g**) assays determining the binding interface between OsPUX8B.2 and OsCDC48-6. **i, j** Co-IP assays showing that OsATG8a (**i**) and OsCDC48-6 (**j**) interact with OsPUX8B.2 but not with OsPUX8B.1 or OsPUX8B.3. **k, l** Competitive Co-IP (**k**) and SLC (**l**) assays showing the effects of MoNLE1 presence on the association between OsPUX8B.2 and OsATG8a or

OsCDC48-6. **m** Time-course analysis of *OsATG8a* and *OsCDC48-6* expression patterns upon blast (strain RB22) infection. Data are means ± s.d. from three biological replicates. Data in **e**, **h** and **l** were shown as boxplots of the number of indicated biological replicates displaying the maximum and minimum, first and third quantiles, and the median. Exact *p* values were determined by two-tailed Student's *t* tests in **l** and **m**. Protein abundance in SLC assays was determined by immunoblot analysis. Note that all Co-IP experiments were performed in rice protoplasts that were pretreated with MG132 for 10 h before collection. Data in **d**, **e** and **g**–**l** are representatives of three independent experiments with similar results.

OsATG8a and OsCDC48-6 specifically interacted with OsPUX8B.2 but not with OsPUX8B.1 or OsPUX8B.3 (Fig. 3i, j), as shown by Co-IP experiments.

The finding that the UBX domain is required for interactions with both OsATG8a and OsCDC48-6 raises the possibility that OsPUX8B.2 might function differently from Arabidopsis PUX8. Moreover, *Ospux8b* mutants were less sensitive to treatment with the CDC48 inhibitor CB-5083 than wild-type plants (Supplementary Fig. 6b), in contrast to their Arabidopsis homologs. We propose that OsPUXB.2 acts as an adapter for either OsATG8 or OsCDC48-6 to target specific substrates for the autophagy- or 26S proteasome-mediated degradation pathway, respectively.

## MoNLE1 promotes the association between OsPUX8B.2 and OsCDC48-6

As the UBX domain of OsPUX8B.2 is required for all known interactions of this protein (e.g., with MoNLE1, OsATG8a, and OsCDC48-6), we asked whether MoNLE1 interferes with the associations between OsPUX8B.2 and OsCDC48-6 or those between OsPUX8B.2 and OsATG8. Accordingly, we performed competitive Co-IP assays in rice protoplasts and established that, surprisingly, MoNLE1 does not clearly affect the binding between OsPUX8B.2 and OsATG8a but enhances the association between OsPUX8B.2 and OsCDC48-6 (Fig. 3k). We confirmed this result using competitive SLC assays (Fig. 3l). Moreover, both OsATG8a and OsCDC48-6 precipitated MoNLE1 in the presence

of OsPUX8B.2 (Fig. 3k), indicating that MoNLE1 binds to OsPUX8B.2 via a interface different from that of OsATG8a and OsCDC48-6. As *OsCDC48-6* expression was drastically and continuously induced, whereas that of *OsATG8a* showed only a minor change upon *M. oryzae* infection (Fig. 3m), we hypothesize that OsCDC48-6 might be specifically required by OsPUX8B.2 to degrade substrates involved in pathogenesis.

We examined attributes of OsCDC48-6 that might be required by OsPUX8B.2 to execute its defense-related functions quickly and efficiently. First, we performed a subcellular localization analysis in rice protoplasts, which showed that, similar to OsPUX8B.2, OsCDC48-6 is distributed in the nucleus and cytoplasm (Supplementary Fig. 6c), suggesting that the OsPUX8B.2–OsCDC48-6 module functions in both compartments. Second, unlike OsCDC48-1, OsCDC48-6 lacks the conserved N domain, which makes it structurally different from typical CDC48 proteins assembling into hexamers, as revealed by homology modeling (Supplementary Fig. 6a, d). This feature may confer OsCDC48-6 increased ATPase activity and affinity for the 26S proteasome, as reported for archaeal CDC48[34,35]. Third, OsCDC48-6 associated with the other five OsCDC48s in addition to itself in SLC assays (Supplementary Fig. 6e, f), which may enable OsCDC48-6 to efficiently participate in the OsPUX8B.2 signaling pathway by coupling with other CDC48 family members, such as OsCDC48-1 and OsCDC48-8, whose encoding genes were co-expressed with *OsPUX8B* (Supplementary Fig. 6g).

### OsPUX8B.2 undergoes 26S proteasome-mediated degradation promoted by MoNLE1

Compared with other splicing variants and OsPUX8A, OsPUXB.2 was highly unstable in rice cells, as we detected very low abundance of OsPUX8B.2-HA by immunoblotting when the encoding construct was driven by the cauliflower mosaic virus (CaMV) 35S promoter (Fig. 4a). By deleting each functional domain, we determined that the UBX domain is responsible for the stability of OsPUX8B.2 (Fig. 4b). To increase the amount of OsPUX8B.2 to a detectable level, we used the maize promoter *Ubipro* to drive the expression of *OsPUX8B.2* cloned in-frame with a sequence encoding an N-terminal *4 × HA* tag (*Ubipro:4 × HA-OsPUX8B.2*) in rice protoplasts, followed by treatment with protease and proteasome inhibitors. The 26S proteasome inhibitor MG132 strongly promoted the accumulation of 4 × HA-OsPUX8B.2, while the serine protease inhibitor phenylmethylsulphonyl fluoride (PMSF) and the autophagy inhibitors 3-methyladdenine (3-MA) and concanamycin A (ConA) did not (Fig. 4c), suggesting that the protein stability of OsPUX8B.2 is mainly regulated by the 26S proteasome. In agreement with this result, an in vivo ubiquitination assay[36,37] showed that OsPUX8B.2 is ubiquitinated in rice cells (Fig. 4d). The CDC48 inhibitor CB-5083 also increased 4 × HA-OsPUX8B.2 protein levels, although to a lesser extent than MG132 treatment (Fig. 4c).

Since the UBX domain is required for OsPUX8B.2 degradation, we reasoned that OsPUX8B.2 might also be regulated by OsCDC48-6-mediated proteasome turnover. Indeed, the abundance of 4 × HA-OsPUX8B.2 in rice protoplasts was lower when its encoding construct was co-transfected along with *OsCDC48-6-GFP*, but not with *GFP* or *GFP-OsATG8a* (Fig. 4e). In addition, the degradation of 4 × HA-OsPUX8B.2 was blocked by treatment with MG132 or CB-5083 (Fig. 4e). This finding points to a possible self-regulatory mechanism of CDC48 adapters; however, it is equally possible that OsPUX8B.2 is degraded during OsCDC48-6-mediated substrate degradation. In the structure of OsPUX8B.2 predicted by AlphaFold, the UBX domain was spatially adjacent to the UBA domain, suggesting functional interplay between domains (Fig. 4f). We propose that this type of domain organization facilitates the processing of substrates by OsPUX8B.2 via OsCDC48-6, which might be tightly regulated by OsPUX8B.2 abundance. On the contrary, OsPUX8B.1,

which did not associate with OsCDC48-6, was not regulated by OsCDC48-6 (Supplementary Fig. 6h). Interestingly, co-expression of *MoNLE1-GFP* with *4 × HA-OsPUX8B.2* in rice protoplasts followed by treatment with the protein translation inhibitor cycloheximide (CHX) resulted in lower OsPUX8B.2 levels than in controls, while co-treatment with CHX and MG132 or CB-5083 resulted in the accumulation of OsPUX8B.2 to levels at least equivalent to those of controls (Fig. 4g). More importantly, we observed the destabilization of MoNLE1 in the presence of OsPUX8B.2 and OsCDC48-6, an effect that was also blocked by MG132 treatment or more weakly by CB-5083 treatment (Fig. 4h). These results are consistent with the finding that MoNLE1 promotes the association of OsPUX8B.2 with OsCDC48-6 (Fig. 3k, l) and suggest that MoNLE1 and OsPUX8B.2 tend to promote each other's depletion though OsCDC48-6–26S proteasome-mediated degradation. Considering that MoNLE1[NES] did not promote OsPUX8B.2 degradation, as shown in Fig. 2h, we speculate that this degradation occurs in plant nuclei.

To explore how the OsPUX8B.2–OsCDC48-6 module participates in the rice–blast interaction, we examined the role of OsCDC48-6 in *M. oryzae* infection. We were not able to generate *OsCDC48-6* knockout plants, which is consistent with the finding that CDC48 is essential for normal plant development[38,39]. Therefore, we generated *OsCDC48-6-HA*-OE plants and examined their phenotypes when challenged by *M. oryzae* inoculation (Supplementary Fig. 6i). We observed that, similar to OsPUX8B.2, OsCDC48-6 also positively regulates the defense response of rice against blast pathogen (Fig. 4i), suggesting that OsCDC48-6 likely functions with OsPUX8B.2 to modulate other substrate proteins rather than directly degrading OsPUX8B.2.

### Proximity-labeling identification of PUX8B.2-associated proteins in rice

To obtain a comprehensive view of the mechanism behind OsPUX8B.2 activity, we used the TurboID-based proximity-labeling method, a powerful tool for identifying protein complexes or protein interaction networks[40–42], to define the complement of OsPUX8B.2-associated proteins. Accordingly, we generated *Ubipro:OsPUX8B.2-TurboID-HA* transgenic plants and performed biotin labeling and mass spectrometry (MS). The resulting MS data were subjected to ratiometric analysis by comparing the normalized peptide intensities between biotin-treated (+) and unlabeled (−) plants (Fig. 5a and Supplementary Fig. 7a, b). Using a *p*-value < 0.05 and a Log₂(FC) > 1 as cutoffs, we identified only seven potential interactors of OsPUX8B.2 (Fig. 5b and Supplementary Data 4). We reasoned that OsPUX8B.2-TurboID-HA might be unstable and maintained at low levels in plants. Since OsPUX8B.2 abundance is tightly regulated by the 26S proteasome, we simultaneously treated the transgenic plants with MG132 and repeated the biotin treatment. In this second round of analysis, we used plants expressing *GFP-TurboID-HA* with or without biotin treatment as controls. We identified 96, 214, and 310 proteins by ratiometric analysis for the comparisons (OsPUX8B.2+biotin) vs. (OsPUX8B.2−biotin), (OsPUX8B.2+biotin) vs. (GFP+biotin), and (OsPUX8B.2+biotin) vs. (GFP−biotin), respectively (Fig. 5c–e, Supplementary Fig. 7c, d, and Supplementary Data 4). Gene Ontology (GO) term enrichment analysis of the proteins present in at least two comparisons indicated that the proteins that potentially associate with OsPUX8B.2 are mainly involved in 'RNA processing and modification,' 'protein folding,' and 'heat/light responses' (Fig. 5f). Fifty-eight proteins were identified by all three comparisons of MG132-treated samples, which we refer to as core preys (CPs) for OsPUX8B.2 (Fig. 5g, h). We predicted that approximately half (28) of these CPs localize to the nucleus, with the other preys localized to the cytoplasm, organelles, and the plasma membrane (Fig. 5i), suggesting that OsPUX8B.2 primarily functions in the nucleus but also has more broad roles.

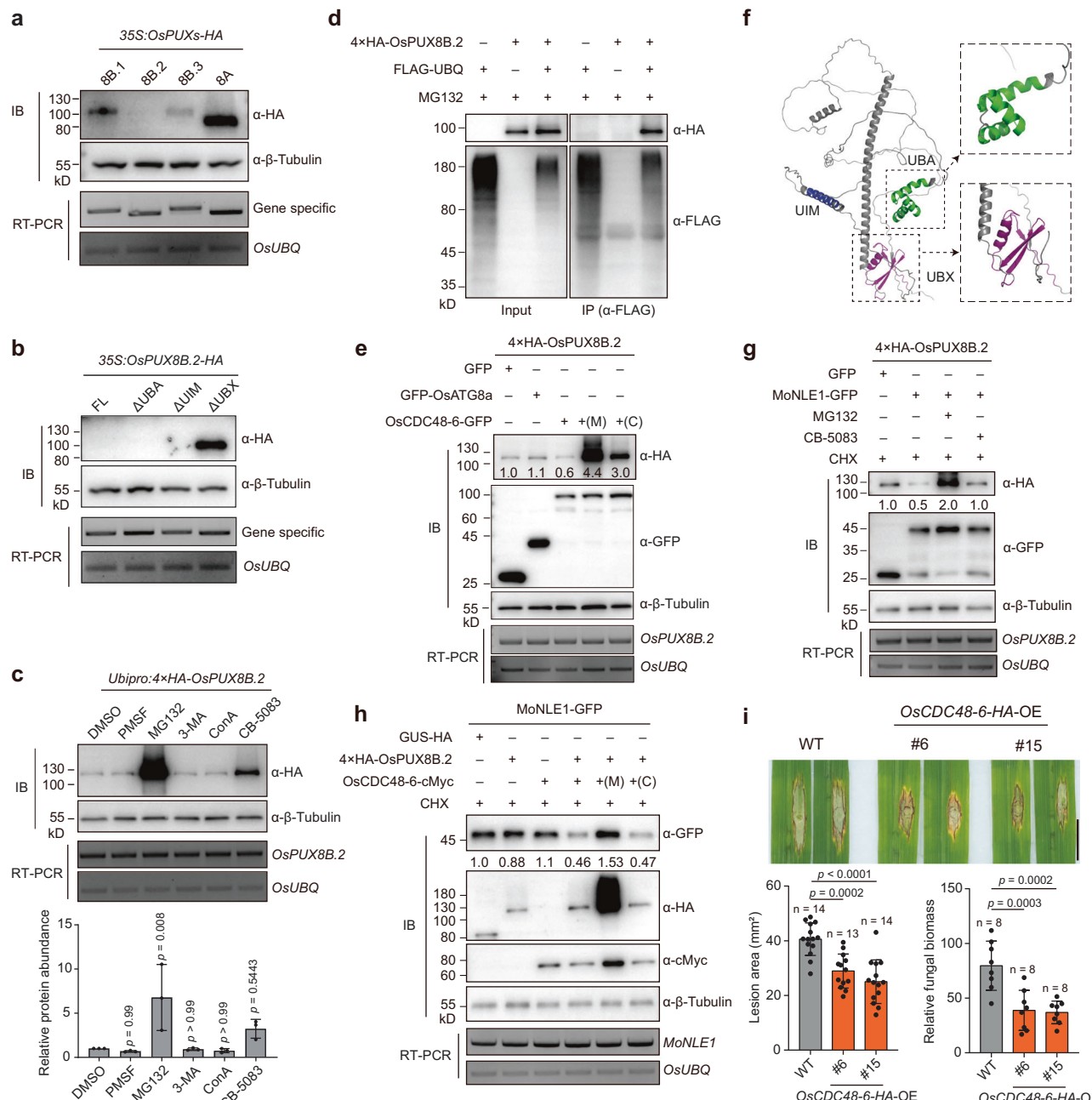

**Fig. 4 | MoNLE1 and OsPUX8B.2 mutually deplete each other through the OsCDC48-6–26S proteasome pathway. a** Immunoblot analysis of PUX proteins in rice protoplasts transfected with the indicated constructs. **b** Immunoblot analysis of domain/motif deletion variants of OsPUX8B.2. **c** Treatment of rice protoplasts expressing *4×HA-OsPUX8B.2* from the maize *Ubiquitin* promoter (*Ubipro*) with protease inhibitors. Each indicated protease inhibitor (with DMSO as a control) was added to the protoplast suspension at 12 h post transfection (hpt) with *Ubipro:4×HA-OsPUX8B.2* and was incubated for 10 h before protein extraction. Relative protein levels of OsPUX8B.2 normalized to β-tubulin were quantified using ImageJ. Data are means ± s.d. from three independent biological replicates. Significant levels (*p* values) were determined with one-way ANOVA followed by Tukey's test by comparing different treatments with the control (DMSO). **d** In vivo ubiquitination assay detecting OsPUX8B.2 ubiquitination in rice protoplasts transfected with the indicated constructs. **e** Degradation of OsPUX8B.2 is promoted by OsCDC48-6 but not by GFP-OsATG8a or free GFP. The indicated constructs were co-transfected into rice protoplasts and detected at 24 hpt. **f** Predicted structure of OsPUX8B.2

obtained from the AlphaFold Protein Structure Database; domains were labeled using PyMOL. **g** OsPUX8B.2 degradation is accelerated by MoNLE1, which is blocked by treatment with MG132 or CB-5083. **h** MoNLE1 is destabilized by OsPUX8B.2, which is strongly and weakly blocked by MG132 and CB-5083, respectively. **i** Overexpressing *OsCDC48-6-HA* in rice enhances plant immunity against blast fungus. Punch inoculation was performed with blast isolate RB22, and disease symptoms were recorded at 10 dpi. Lesion area and relative fungal biomass were used to quantify disease severity. Data are means ± s.d. from the indicated number of independent biological replicates. Adjusted *p* values were determined by one-way ANOVA followed by Tukey's test. Scale bar, 1 cm. In **a**–**c**, **e**, **g**, and **h**, β-tubulin was used as a loading control; numbers below the protein bands **e**, **g** and **h** indicate the relative abundances of corresponding proteins compared to β-tubulin; and RT-PCR analysis was used to assess the transcript levels of the corresponding genes with *OsUBQ* as an internal control. Data in **a**, **b**, **d**, **e**, **g**, **h** are representatives of three independent experiments with similar results.

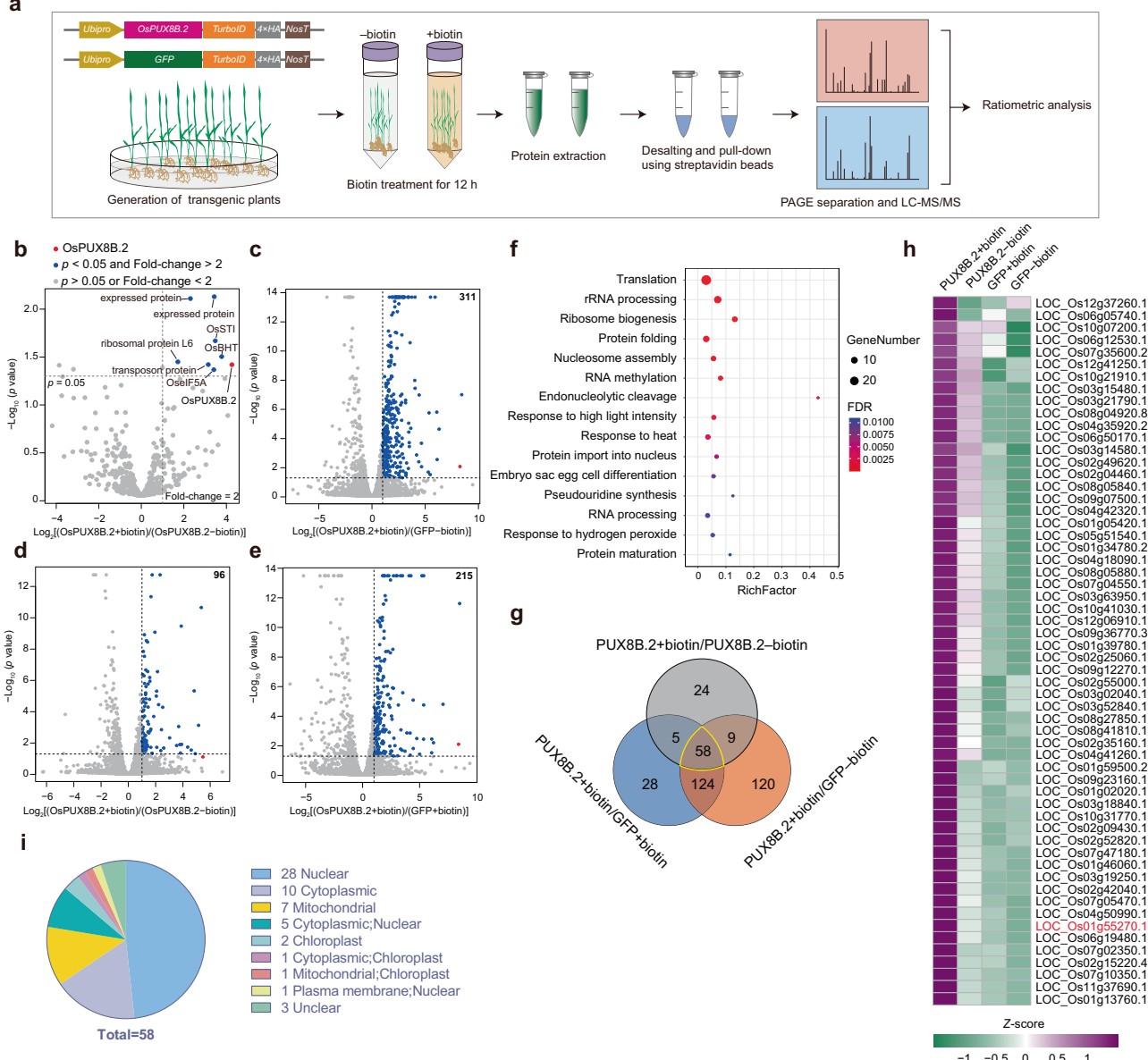

**Fig. 5 | Proximity labeling with TurboID identified proteins interacting with OsPUX8B.2 in rice. a** Workflow of proximity-labeling spectrometry (PL-MS/MS) using rice seedlings harboring the indicated *TurboID*-fusion constructs. **b–e** Ratiometric analysis of OsPUX8B.2 preys identified by PL-MS/MS. The protein enrichment areas (LFQ intensities) were used for all quantitative ratiometric analyses. The cutoffs of adjusted *p*-value < 0.05 and Log₂(FC) > 1 were used for selection of putative preys of OsPUX8B.2. In **b**, non-MG132-treated samples were analyzed by comparing biotin-treated (+biotin) data with non-biotin-treated (−biotin) data. In **c–e**, all seedlings were treated with MG132, and the MS data were

quantified by comparing the OsPUX8B.2+biotin data with the OsPUX8B.2−biotin (**c**), GFP+biotin (**d**), and GFP−biotin (**e**) data. **f** Gene Ontology (GO) term enrichment analysis of proteins present in at least two groups from the quantitative analysis in **c–e**. **g** Venn diagram showing the extent of overlap between proteins identified in **c–e**. The 58 core preys (CPs) are labeled in yellow. **h** Relative protein abundance of the 58 CPs. Relative protein levels (Log₂LFQ) in MS data from three biological replicates were used for the analysis. Data were normalized per row for each protein using TBtools[76]. The protein OsBHT (encoded by LOC_Os01g55270.1) is labeled in red. **i** Predicted subcellular localizations of the 58 CPs.

## Identification of OsBHT as a putative cytoplasmic substrate of OsPUX8B.2

Among the 58 CPs, we repeatedly identified one protein encoded by LOC_Os01g55270 in all ratiometric analysis comparisons, even in samples not treated with MG132 (Fig. 5b–e, h). This protein, named BOOTING STAGE HIGH-TEMPERATURE TOLERANCE (OsBHT), is involved in heat tolerance as a cochaperone (Supplementary Fig. 7e)[43]. Notably, similar to *OsPUX8B* and *OsCDC48-6*, the transcript level of *OsBHT* was significantly induced during *M. oryzae* infection (Supplementary Fig. 7f), suggesting that OsBHT might be involved in the rice–*M. oryzae* interaction. We therefore explored the functional relationship of OsBHT with OsPUX8B.2.

In a Co-IP assay, OsPUX8B.2 showed a robust interaction with OsBHT (Fig. 6a). Moreover, similar to MoNLE1, OsATG8a, and OsCDC48-6, OsBHT specifically interacted with OsPUX8B.2, but not with OsPXU8B.1 or OsPUX8B.3 (Fig. 6b). However, the binding of OsBHT required both the UBA and UBX domains of OsPUX8B.2 (Fig. 6c), which was different from that of MoNLE1, OsATG8a, and OsCDC48-6. OsBHT was predominantly located in the cytoplasm, where it co-localized with OsPUX8B.2 (Fig. 6d). Although OsBHT is thought to function as a cochaperone, it showed a moderate effect on the stability of OsPUX8B.2 (Supplementary Fig. 7g). Moreover, an in vivo ubiquitination assay suggested that OsBHT can be bound by ubiquitin in rice cells (Fig. 6e). These results suggest that OsBHT might

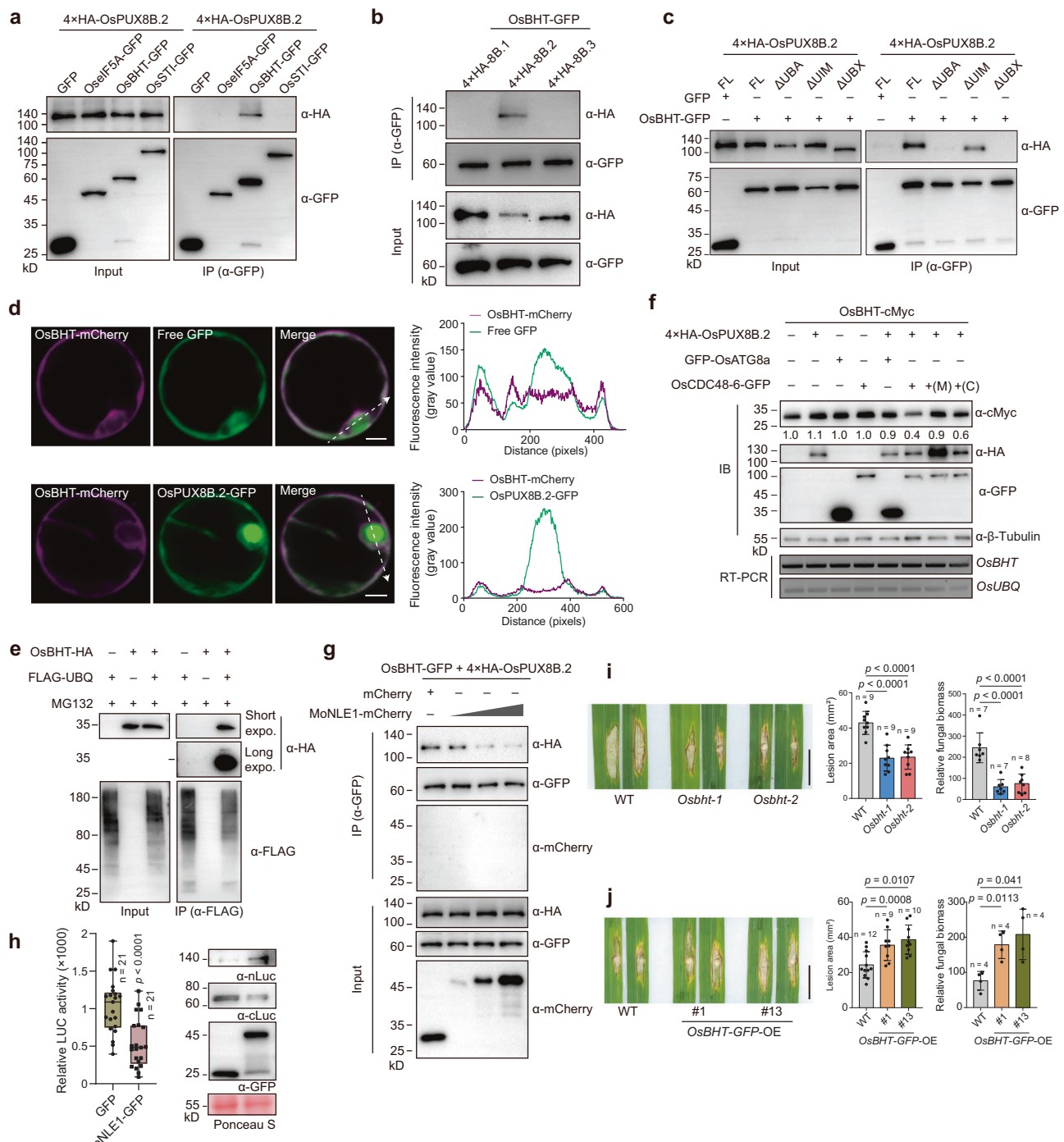

**Fig. 6 | OsBHT is a putative substrate of OsPUX8B.2–OsCDC48-6. a** Co-IP assay showing that OsPUX8B.2 associates with OsBHT but not with OseIF5A or OsSTI. **b** Co-IP assay showing that OsBHT only binds to OsPUX8B.2 but not to OsPUX8B.1 or OsPUX8B.3. **c** Co-IP assay showing that the association between OsBHT and OsPUX8B.2 requires the UBA and UBX domains of OsPUX8B.2. **d** Colocalization of OsBHT-mCherry with free GFP (upper panel) or OsPUX8B.2-GFP (lower panel). Fluorescence intensities (gray values) across the white lines are shown in the line plots on the right. **e** In vivo ubiquitination assay showing that OsBHT is ubiquitinated in rice protoplasts transfected with the indicated constructs. **f** Immunoblot analysis of proteins extracted from rice protoplasts showing that OsPUX8B.2–OsCDC48-6 promotes the turnover of OsBHT. Numbers below the protein bands indicate the relative abundances of OsBHT-cMyc compared to β-tubulin. **g, h** Competitive Co-IP (**g**) and

SLC (**h**) assays showing that MoNLE1 disrupts the association between OsBHT and OsPUX8B.2. **i, j** Phenotypes of *Osbht* mutants (**i**) and *OsBHT-GFP*-OE plants (**j**) infected with blast fungus. Punch inoculation was performed with blast isolate RB22, and the disease symptoms were recorded at 10 dpi. Lesion area and relative fungal biomass were used to quantify disease severity. Scale bars, 1 cm. Data were shown as boxplot of the number of indicated biological replicates displaying the maximum and minimum, first and third quantiles, and the median (**h**); or means ± s.d. from the indicated number of independent biological replicates (**i, j**). Data were analyzed by two-tailed Student's *t*-tests (**h**) or one-way ANOVA followed by Tukey's test (**i, j**). Exact *p* (**h**) or adjusted *p* (**i, j**) values were shown in figures. Data in **a–g** are representatives of three independent experiments with similar results.

be a cytoplasmic substrate of OsPUX8B.2. To test this hypothesis, we co-expressed *OsBHT*, *OsPUX8B.2*, and *OsCDC48-6* or *OsATG8a* in rice protoplasts. OsBHT was efficiently degraded in the presence of OsPUX8B.2 and OsCDC48-6, but not with OsPUX8B.2 alone or with OsPUX8B.2 and OsATG8a; this degradation was effectively blocked by MG132 or CB-5083 treatment (Fig. 6f). Moreover, upon CHX treatment, OsBHT-GFP showed a lower protein abundance in protoplasts prepared from *OsPUX8B.2-HA*-OE plants than from wild-type plants, while its abundance differed little in protoplasts prepared from *OsPUX8B.1-HA*-OE plants (Supplementary Fig. 7h). These results indicate that OsBHT is a high-confidence candidate substrate of the OsPUX8B.2–OsCDC48-6 module. Moreover, MoNLE1 was capable of disrupting the association between OsBHT and OsPUX8B.2, as determined by competitive Co-IP and SLC assays (Fig. 6g, h), further supporting the notion that OsBHT participates in the pathogenesis of *M. oryzae*.

To explore the role of OsBHT in blast infection, we generated *Osbht-ko* mutants using CRISPR/Cas9-mediated gene editing, as well as *OsBHT-GFP*-OE plants (Supplementary Fig. 7i, j). Inoculation assays suggested that *Osbht-ko* plants exhibited increased resistance against *M. oryzae* infection, whereas *OsBHT-GFP-OE* plants displayed the opposite phenotype, suggesting that OsBHT is a negative regulator of blast disease resistance (Fig. 6i, j). As OsPUX8B.2 and OsCDC48-6 positively regulate plant immunity against blast infection, we suggest that OsPUX8B.2 functions by modulating OsBHT levels, but this effect might be overcome by the *M. oryzae* effector MoNLE1.

## Discussion

In this study, we explored the host protein targeted by the core fungal effector MoNLE1, which unveiled the anti-blast function of the PUX protein OsPUX8B.2 and its cooperation with OsCDC48-6 to mediate the turnover of OsBHT. MoNLE1 is a unique orphan effector of the fungus *M. oryzae* that is not present in other fungal species. Its high sequence similarity in all tested blast isolates from rice confirms its role as a core effector for rice blast strains. The adaptation of *M. oryzae* to rice as a host is believed to have been facilitated by a few lineage-specific genes[44], and the greater number of gene losses compared with other isolates indicates a longer evolutionary history of host specificity for rice isolates[45]. Therefore, we hypothesize that during *M. oryzae* evolution, lineage-specific *MoNLE1* was selectively advantageous and played a critical role in the adaptation of *M. oryzae* to rice. However, this adaptation does not appear to be associated with the targeting of OsPUX8B.2, as this function is highly conserved among MoNLE1 homologs.

Although the *OsPUX8B* locus produces three splice variants that each encode a slightly different protein, only the OsPUX8B.2 isoform showed binding to all identified interactors in this study. The protein abundance of OsPUX8B.2 is tightly regulated by the 26S proteasome, and the transcript level of *OsPUX8B.2* is maintained at a low level under normal growth conditions (Fig. 4c and Supplementary Fig. 5c), indicating that OsPUX8B.2 must be tightly controlled to avoid the misregulation of cellular events. OsPUX8B.2 binds to rice ATG8 proteins and specifically associates with OsCDC48-6. However, unlike its Arabidopsis homologs[22], the UBX domain but not the UIM of OsPUX8B.2 is required for the association of this protein with both OsATG8s and OsCDC48-6 (Fig. 3c–h), which strengthens the notion that the UIM is not always required for binding to ATG8[22] and suggests that OsPUX8B.2 might function differently from AtPUX8. We propose that OsPUX8B.2 functions as an adapter for either OsATG8 or OsCDC48-6, which is determined by specific signaling events, such as pathogen infection.

We also identified OsBHT as a putative substrate of the OsPUX8B.2–OsCDC48-6 module (Figs. 5 and 6). *OsBHT* is reported to contribute to high-temperature tolerance in rice at the booting stage[43]. Signature domains of OsBHT include the CS (CHORD [cysteine- and

histidine-rich domain]-containing) and SGS (SGT1-specific) domains, which are present in SUPPRESSOR OF THE G2 ALLELE OF SKP1 (SGT1), a key regulator of multiple plant resistance proteins such as nucleotide-binding leucine-rich repeat (NLR) receptors[46,47]. Whether OsBHT is involved in NLR-mediated resistance is currently unknown; however, our data indicate that OsBHT plays negative roles in plant basal defense against blast pathogen (Fig. 6h, i). *OsBHT* expression is induced by *M. oryzae* to facilitate pathogen infection (Supplementary Fig. 7f), while as a counterattack strategy, OsPUX8B.2–OsCDC48-6 promotes the degradation of OsBHT to sustain a proper immune level in the plant (Fig. 7a). However, the opposite phenotypes observed in *OsPUX8B.1-HA*-OE and *OsPUX8B.2-HA*-OE plants following blast infection cannot simply be explained by OsBHT function, as OsPUX8B.1 does not affect OsBHT binding or stability (Fig. 6b and Supplementary Fig. 7i). Whether other proteins associate with both MoNLE1 and all three OsPUX8B splicing isoforms and participate in immune responses needs to be determined in future studies.

In conclusion, we uncovered the function of MoNLE1 in disrupting the rice OsPUX8B.2–CDC48-6–OsBHT signaling pathway. Based on our observations, the pathogenic virulence factor MoNLE1 has at least three effects on OsPUX8B.2. First, MoNLE1 binds to OsPUX8B.2 and promotes its nuclear migration; second, MoNLE1 destabilizes OsPUX8B.2 by increasing its association with OsCDC48-6, which also causes the turnover of MoNLE1 itself; and third, MoNLE1 depletes OsBHT from OsPUX8B.2, which might help maintain OsBHT protein levels (Fig. 7b). All three MoNLE1 functions described above weaken

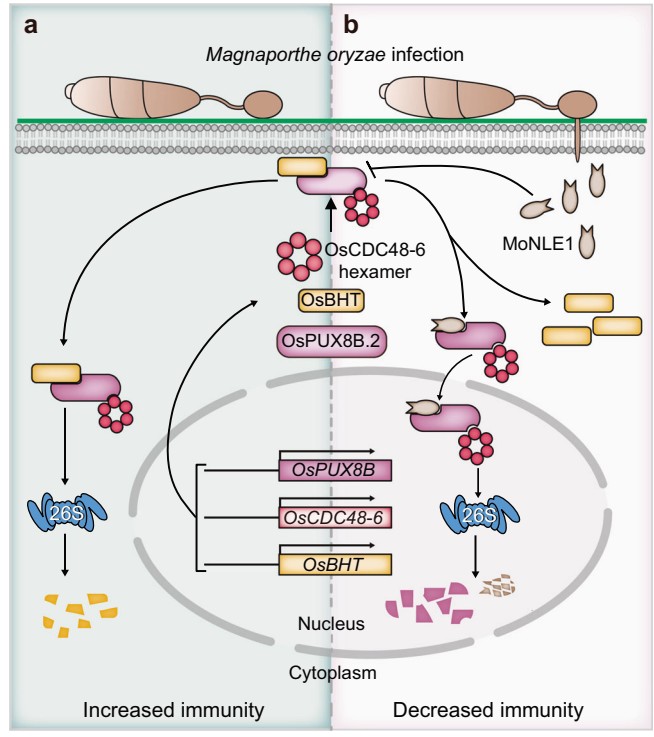

**Fig. 7 | A model for MoNLE1 disturbing OsPUX8B.2 function. a** Upon *M. oryzae* infection, the expression of *OsPUX8B.2*, *OsCDC48-6*, and *OsBHT* is induced. OsBHT negatively regulates rice defense against *M. oryzae* infection, while the OsPUX8B.2–OsCDC48-6 module promotes degradation of OsBHT to maintain a proper immune level in plants. **b** *M. oryzae* has evolved the conserved core nuclear effector MoNLE1 to counteract the role of OsPUX8B.2–OsCDC48-6 in regulating host immunity. First, MoNLE1 directly binds to OsPUXB.2 and promotes its migration to the nucleus; second, binding of MoNLE1 to OsPUX8B.2 facilitates the association of OsPUX8B.2 with OsCDC48-6 and stimulates the degradation of OsPUXB.2 and MoNLE1 in rice nuclei; third, MoNLE1 disrupts the connection between OsPUX8B.2 and OsBHT, which may help stabilize OsBHT and thus support a successful invasion of *M. oryzae*.

host immune responses and support *M. oryzae* infection. Nevertheless, our results also shed light on the possibility of targeting MoNLE1 by manipulating its host target proteins. Moreover, considering the high conservation of MoNLE1 in blast strains, rice plants with broad, durable blast resistance might be achieved by engineering intracellular MoNLE1 resistance proteins in plants.

## Methods

### Plant materials and growth conditions

The highly blast-susceptible rice (*Oryza sativa*) cultivar CO39 was used to determine the pathogenicity of *monle1* and the complementation strains. All transgenic plants and mutants used in this study were generated in the Nipponbare (NPB) background via Agrobacterium (*Agrobacterium tumefaciens*)–mediated transformation of rice callus with the corresponding constructs. $T_0$ plants selected by corresponding antibiotic were further screened by PCR, Sanger sequencing, or RT-qPCR. All rice plants were cultivated in a greenhouse or field for seed propagation. Surface-sterilized seeds were germinated on solid half-strength Murashige and Skoog medium for 7 d, and the seedlings were transplanted to soil for continued growth. *Magnaporthe oryzae* punch inoculation was performed on 5- to 6-week-old rice plants grown in a growth chamber under the following conditions: 25 °C, 80% relative humidity, and a 12-h-light/12-h-dark photoperiod.

*Nicotiana benthamiana* plants were grown on soil in a greenhouse at 22 °C, 60% relative humidity, and a 14-h-light/10-h-dark photoperiod.

### Blast fungus and growth conditions

The *M. oryzae* isolates GUY11 and RB22 were used in this study. The knockout strain for *MoNLE1* and the complementation strains were generated in the GUY11 background. RB22 was used to determine the phenotypes of the transgenic plants. The titer of *M. oryzae* spores was adjusted to $1.5 \times 10^5$/mL and $3 \times 10^5$/mL for spray and punch inoculation, respectively. Inoculation was performed as previously described[33]. All *M. oryzae* stains were cultivated on oat medium plates (3% [w/v] milled oats and 1.5% [w/v] agar) at 25 °C under constant light.

### Gene cloning and plasmid construction

Because there is no intron in the genome sequence of *MoNLE1*, the full-length protein-coding sequence (CDS) of *MoNLE1* was amplified from genomic DNA of the *M. oryzae* strain GUY11. Unless otherwise indicated, experiments were performed using a *MoNLE1* sequence encoding a protein lacking the N-terminal targeting signal. To obtain the CDSs of rice genes, total RNA was extracted from NPB plants using an RNA Isolation kit (FOREGENE, Cat# RE05014) and reverse-transcribed to first-strand cDNA using a First Strand cDNA Synthesis kit (Vazyme, Cat# R212-01). Following PCR amplification using high-fidelity polymerase (TOYOBO LIFE SCIENCE, Cat# KFX-101), the CDSs were cloned into the corresponding vectors using a One Step Cloning Kit (Vazyme, Cat# C112) or T4 ligase (Thermo Scientific, Cat# EL0012). For domain deletion/mutation variants, the corresponding DNA fragments were cloned by overlapping PCR.

Specifically, for generation of the *MoNLE1^NES^* construct, the sequence encoding the nuclear export signal (NES) from HIV-1 Rev protein[48] was used to replace the sequence encoding the nuclear localization signal (NLS) of MoNLE1. To generate rice plants overexpressing individual genes, genes were cloned into the pRHVcHA vector harboring the maize *Ubiquitin* promoter (*Ubipro*) and a sequence encoding a C-terminal HA tag[49]. The *Ubipro:OsPUX8B.2-TurboID-HA* and *Ubipro:GFP-TurboID-HA* constructs were generated by sequentially subcloning the *TurboID* and *OsPUX8B.2* or *GFP* CDSs into the pRHVcHA vector. To generate knockout plants via CRISPR/Cas9-mediated gene editing, a single guide RNA (sgRNA) was designed using CRISPR-P 2.0 (http://crispr.hzau.edu.cn/CRISPR2/) and inserted in the pEntryA vector. *U6P1:sgRNA* was removed from pEntryA by digestion and cloned into pRHvCas9[49].

Primers used to generate the constructs are listed in Supplementary Data 5.

### Effector prediction and conservation pattern analysis

A systematic approach was employed to identify core effectors conserved in *Oryza* pathotype strains, as shown in Supplementary Fig. 1b. The CDSs of predicted effectors were then used as queries for BLAST searches against 187 high-quality *M. oryzae* genome assemblies to determine their presence/absence variation (PAV) in each strain. A threshold of at least 95% identity, 85% coverage, and hit counts lower than the exon count were applied for identifying effector presence in a given strain. Based on the resulting PAV matrix, an effector was considered highly conserved in the *Oryza* pathotype if it was present in at least 110 out of the total 113 *Oryza* pathotype strains and absent in at least 70 out of 74 strains representing other pathotypes.

### Phylogenetic analysis

To conduct gene-based phylogenetic analysis of *MoNLE1*, the genomic sequences of a comprehensive collection of 187 *M. oryzae* isolates from 13 host genera were acquired from previous reports[50–63]. The analysis included sequence data from the genome assemblies of strains collected from rice, wheat, and grasses (Supplementary Data 2). A distance-based neighbor-joining tree was reconstructed with R v4.4.0 using the packages adegenet v2.1.3[64], ape v5.4[65], and ggtree v2.2.4[66]. The Saitou and Nei method of neighbor-joining estimation was performed with the "nj" function of ape and plotted using ggtree. BLASTN v2.9.0[60] was used to determine the presence or absence of an insertion within *MoNLE1* and to calculate sequence similarity within each clade. A synteny plot was generated using the MCScanX algorithm by comparing the genomic arrangement of three isolates from *Oryza*, *Triticum*, and *Eleusine* over a 1-Mb window centered on the *MoNLE1* locus.

Rice PUX proteins were predicted by a HMMER search against the rice (*Oryza sativa* ssp. *japonica*) protein database using the UBX domain (PF00789) as a query, which returned 13 candidate proteins. One protein (encoded by LOC_Os10g22570.1) was excluded by manual inspection due to the high *e*-value of its UBX domain. The remaining PUX family members were named according to their similarity to their Arabidopsis homologs. The sequences of 16 Arabidopsis PUX proteins were retrieved from The Arabidopsis Information Resource database (https://www.arabidopsis.org/). A BLASTp search was used to identify homologs of OsPUX8B in 10 other members of the green lineage: *Medicago truncatula*, *Brachypodium distachyon*, *Sorghum bicolor*, *Zea mays*, *Glycine max*, *Triticum aestivum*, *Selaginella moellendorffii*, *Nicotiana attenuata*, *Arabidopsis thaliana*, and *Ostreococcus lucimarinus*. To identify rice CDC48 family members, the yeast (*Saccharomyces cerevisiae* S288C) CDC48 (GenBank accession: NP_010157.1) sequence was used as a query to search the rice protein database with BLASTp. Rice CDC48s were named OsCDC48-1 to -8 based on their similarity to yeast CDC48. Phylogenetic trees were reconstructed using the neighbor-joining method in MEGA7 (v7.0) with the indicated protein sequences. Numbers next to branches indicate the percentage of replicate trees in which the associated taxa clustered in a bootstrap test (1,000 replicates). Scale bars represent the number of amino acid substitutions per site. Protein domains or motifs were predicted using Pfam or by alignment with homologs in other plants. All protein sequence information can be found in Supplementary Data 3.

### Generation and complementation of rice blast mutants

To generate knockout mutants of *MoNLE1*, the -0.8-kb upstream and downstream flanking fragments of *MoNLE1* were cloned into the vector PKOV2.1, which contains a hygromycin phosphotransferase (*HPT*) gene. The DNA fragment comprising the upstream and downstream flanking fragments and *HPT* was amplified by PCR using high-fidelity polymerase (TOYOBO LIFE SCIENCE, Cat# KFX-101) and transformed into *M. oryzae* strain GUY11 as described previously[67]. Transformants were identified

using three pairs of primers: forward, upstream of fragment A (UA) + reverse, *HPT* gene primer (H853); forward, *HPT* gene primer (H708) + reverse, downstream of fragment B (DB); and the *MoNLE1*-specific primers NF/NR. Complementation was performed by transforming the *monle1* mutant with the pFL2 construct carrying *MoNLE1* and the upstream 975-bp promoter (*MoNLE1pro:MoNLE1*) sequence.

## Blast inoculation and phenotypic analysis

To determine the pathogenicity of *monle1* and the complementation strains, 3-week-old CO39 plants were subjected to spray inoculation with $1.5 \times 10^5$/mL fungal spores. For phenotypic analysis of transgenic plants in the NPB background, punch inoculation was performed as described previously[33]. The effects of spray inoculation were measured by counting the number of lesions within 5-cm leaf tips. Lesion area following punch inoculation was measured with ImageJ, and the relative fungal biomass was quantified by genomic DNA–based qPCR as previously described[68]. The *M. oryzae* transposable element *MoPot2* (MGG_13294.6) was used as an indicator of relative fungal biomass, and the rice *UBIQUITIN* gene (*OsUBQ*, LOC_Os03g13170) served as an internal control. qPCR primers for *MoPot2* and *OsUBQ* can be found in Supplementary Data 5.

## Measuring ROS accumulation

Leaf disks from 5-week-old *Ubip:MoNLE1-HA* transgenic (line #6) and wild-type (NPB) plants were immersed in distilled water and incubated overnight at room temperature. Following treatment with elicitor, ROS generation was monitored via a luminol chemiluminescence assay. In detail, three leaf disks per sample were placed in a 1.5-mL microcentrifuge tube containing 100 μL luminol solution (Bio-Rad ImmunStar horseradish peroxidase substrate Cat# 170-5040). Immediately after adding 1 μL of horseradish peroxidase (Jackson ImmunoResearch) and elicitor (100 nM flg22 or 8 μM chitin, or distilled water as a control) to the tube, luminescence was measured in 10-s intervals for 20 min with a Glomax 20/20 luminometer (Promega). Two to three biological replicates were measured per treatment.

## Yeast trap secretion assay

To determine whether MoNLE1 can be secreted, a yeast trap assay was performed as previously described[31]. Briefly, the full-length CDS of *MoNLE1* (*MoNLE1-FL*) or a variant encoding MoNLE1 lacking the signal peptide (*MoNLE1$^{\Delta SP}$*), the positive control *Avr1b-FL* (encoding an effector gene from *Phytophthora sojae*, GenBank accession number AAM20936), and the negative control encoding Mg87 (GenBank accession number XP_362169.1) were separately cloned into the pSUC2 vector (carrying an invertase gene) and used to transform the invertase-negative yeast strain YTK12 using an Alkali-Cation Yeast Transformation Kit (MP, Cat# 2200200). After transformation, yeast cells were selected on synthetic defined (SD)−Trp medium. Positive transformants were confirmed by PCR using gene-specific primers and replicated onto SD−Trp medium with 2% (w/v) sucrose and 2 μg/mL antimycin A to identify pSUC2-postive cells and on YPRAA medium (1% [w/v] yeast extract, 2% [w/v] peptone, 2% [w/v] raffinose, and 2 μg/mL antimycin A) to identify invertase-secreting cells. The chemical reagent 2,3,5-triphenyltetrazolium chloride (TTC) was used to monitor invertase activity by observing the degree of color change. Yeast cells were resuspended in 5 mL SD−W or YPDA (for untransformed YTK12 cells) medium for 24 h at 30 °C and collected by centrifugation (1000 × *g*, 2 min) at room temperature, and the pellet was washed three times with distilled water and incubated in 0.1% (w/v) colorless TTC at 35 °C for 35 min. A color change of the solution from colorless to red indicates the secretion of the corresponding protein.

## Subcellular localization analysis

To examine the movement of MoNLE1 during blast infection, the 519-bp full-length CDS of *MoNLE1* was amplified and inserted into the vector pGTN to generate the *RP27pro:MoNLE1-GFP* construct. The construct was introduced into the rice blast strain GUY11 by the protoplast transformation method[69]. Strains identified as positive by PCR were used for dip inoculation of 6-week-old rice leaf sheaths (cultivar CO39) at a titer of $1.5 \times 10^5$ spores/mL. Inoculated rice sheaths were incubated in a dark, humid chamber at 25 °C, and the epidermal cells of infected rice sheaths were observed immediately after inoculation and at 30 and 42 h post inoculation (hpi) under a Leica DM6 B fluorescence microscope. Images were captured using LAS X software (Leica Microsystems).

Subcellular localization was performed by transfecting NPB protoplasts with the corresponding constructs and observing them at 18–24 h post transfection (hpt). Fluorescent signals were captured under a Zeiss LSM880 confocal laser scanning microscope. Wavelengths for excitation and collection of fluorescent protein signals were as follows: GFP was excited at 488 nm and collected at 490–550 nm, YFP was excited at 514 nm and collected at 519–620 nm, and mCherry was excited at 561 nm and collected at 575–640 nm. Images were captured and processed using ZEN lite black and blue editions, respectively.

## Yeast two-hybrid analysis

The MATCHMAKER GAL4 Two-Hybrid System (Clontech) was used for library screening and subsequent verification according to the manufacturer's instructions. The yeast library was generated using cDNA prepared with total RNA extracted from *M. oryzae*–inoculated rice seedings[28]. For targeted verification, full-length or fragments of CDSs were cloned in the pGBKT7 (BD) or pGADT7 (AD) vector as indicated and co-transformed into yeast strain AH109, followed by screening on SD double dropout medium (SD/−Leu−Trp, DDO) for 3 d at 30 °C. Positive colonies were selected and spotted onto DDO medium (as growth control), triple dropout medium (SD/−Leu−Trp−His, TDO), and quadruple dropout medium (SD/−Leu−Trp−His−Ade, QDO) to assess interactions. Results were recorded after 3–5 d of cultivation at 30 °C. Proteins were extracted from yeast cells growing on DDO plates and detected with antibodies recognizing HA (ani-HA, Roche, Cat# 11867423001, dilution 1:5000) and cMyc (ani-cMyc, GenScript, Cat# A00704, dilution 1:5,000) epitope tags present in the AD and BD constructs

## Co-immunoprecipitation assay

All Co-IP experiments were performed in protoplasts from NPB plants. Ten-day-old seedlings cultivated on half-strength Murashige and Skoog medium were used to prepare protoplasts. The protoplasts were transfected with the indicated constructs for protein production and collected at 20–24 hpt for protein extraction. Note that the protoplasts were treated with 50 μM MG132 for 10 h before collection. Total protein was extracted from the samples using native protein buffer (50 mM Tris-MES pH 8.0, 0.5 M sucrose, 1 mM MgCl$_2$, 10 mM EDTA, 5 mM DTT, 50 μM MG132, 0.5% [v/v] Triton X-100, 0.5% [v/v] NP-40, and protease inhibitor cocktail [Roche, Cat# 04693159001, dilution 1 tablet/10 mL]). Immunoprecipitation was performed by incubating the protein samples with GFP-Trap Agarose beads (Chromotek, Cat# gta-20) overnight; beads carrying protein complexes were washed three or four times with phosphate-buffered saline (PBS). Proteins were eluted by boiling the beads for 10 min and separated by SDS-PAGE. The proteins were transferred onto a PVDF membrane and probed by immunoblotting with anti-HA (Roche, Cat# 11867423001, dilution 1:5000), anti-mCherry (EASYBIO, Cat# BE2026, dilution 1:5000), and anti-GFP (EASYBIO, Cat# BE2001, dilution 1:5,000) antibodies. For competitive Co-IP assays, constructs encoding the competitors were transfected separately into protoplasts for protein extraction and added to the Co-IP mixture in a gradient. Proteins were detected with the corresponding antibodies.

## Nuclear fractionation assay

Nuclear fractionation was performed as previously described[70] with minor modifications. Briefly, rice protoplasts producing the corresponding proteins were collected by centrifugation (1,000 × g, 5 min) at room temperature and resuspended in 300 μL lysis buffer (20 mM Tris-HCl pH 7.5, 25% [v/v] glycerol, 20 mM KCl, 2 mM EDTA, 2.5 mM MgCl₂, 250 mM sucrose, 5 mM DTT, and protease inhibitor cocktail). After incubation on ice for 15 min, a 50-μL sample was used as a total protein aliquot; the remaining sample was centrifuged at 1,500 × g for 10 min at 4 °C for isolation of nuclear proteins. The pellet was washed five times with 1 mL NRBT buffer (20 mM Tris-HCl pH 7.5, 25% [v/v] glycerol, 2.5 mM MgCl₂, 0.2% [v/v] Triton X-100, and protease inhibitor cocktail), resuspended in 500 μL NRB2 buffer (20 mM Tris-HCl pH 7.5, 250 mM sucrose, 10 mM MgCl₂, 0.5% [v/v] Triton X-100, 5 mM β-mercaptoethanol, and protease inhibitor cocktail), and carefully overlaid on top of NRB3 buffer (20 mM Tris-HCl pH 7.5, 1.7 M sucrose, 10 mM MgCl₂, 0.5% [v/v] Triton X-100, 5 mM β-mercaptoethanol, and protease inhibitor cocktail). After centrifugation at 16,000 × g for 45 min at 4 °C, the pellet (nuclei) was resuspended in 100 μL lysis buffer. β-Tubulin (EASYBIO, Cat# BE0025, dilution 1:5,000) and Histone H3 (EASYBIO, Cat# BE3015, dilution 1:5,000) were used as quality controls for total and nuclear proteins, respectively.

## Chemical treatment

Rice protoplasts transfected with *Ubipro:4 × HA-OsPUX8B.2* were treated with DMSO (as a control, 0.5% [v/v]), 1 mM PMSF, 5 mM 3-MA, 1 μM ConA, 50 μM MG132, or 50 μM CB-5083 at 12 hpt and incubated for 10 h before collection and isolation of protein for immunoblot analysis.

For treatment of rice seedlings, CB-5083 was added to half-strength MS medium at concentrations of 2.5, 5, 10, or 20 μM with DMSO (0.2%, v/v) as a control. Seven-day-old seedlings were used for phenotyping.

## In vivo ubiquitination assay

In vivo ubiquitination assays in protoplasts were performed as previously described[71]. Briefly, a construct encoding FLAG-tagged Ubiquitin (Flag-UBQ) was co-transfected with the indicated construct encoding an HA-tagged protein in rice protoplasts and pretreated with MG132 for 10 h before protein extraction. Total protein was collected using native protein buffer and subjected to IP assay with Anti-Flag Affinity Gel (GNI, Cat# GNI4510-FG). Proteins were detected with horseradish peroxidase (HRP)−conjugated anti-HA (EASYBIO, Cat# BE7002, dilution 1:3,000) and anti-Flag (Beyotime, Cat# AF2855, dilution 1:3,000) antibodies.

## Split-luciferase complementation (SLC) assay

SLC experiments were performed on *Nicotiana benthamiana* leaves as previously described[28]. Briefly, Agrobacterium cultures carrying corresponding constructs were resuspended in infiltration buffer (10 mM MgCl₂, 10 mM MES pH 5.6, and 200 μM acetosyringone) to a final cell density of OD₆₀₀ = 0.6 and infiltrated into *N. benthamiana* leaves. Forty-eight hours after infiltration, the infiltrated leaves were sprayed with 1 mM D-luciferin (BioVision, Cat# 7903) and LUC activity was immediately recorded using a CCD imaging system (Tanon 5200). Relative luciferase activity was measured using a GloMax 96 Microplate Luminometer (Promega) by immersing two leaf disks (28 mm²/disc) in 150 μL 1 mM luciferin. For competitive SLC assays, Agrobacterium cell suspensions carrying the *GFP* or *MoNLE1-GFP* plasmid were co-infiltrated with the indicated SLC constructs. To avoid possible degradation of OsPUX8B.2 by MoNLE1, *N. benthamiana* leaves producing MoNLE1 were pretreated with MG132 for 10 h before luminescence detection.

## Co-expression analysis

For co-expression analysis of *OsPUX*s and *OsCDC48*s, their logistic mutual ranks (LSMRs) were calculated in ATTED-II (http://atted.jp) using microarray-based expression data. Total LSMR values were displayed as a heatmap using R. The LSMRs between *OsPUX8B* and related genes were displayed as a co-expression network map using Cytoscape, in which the edge weights represent the LSMRs.

## RT-qPCR analysis

Total RNA was extracted from the indicated *M. oryzae* or rice tissues using a Plant Total RNA Isolation Kit (FOREGENE, Cat# RE05014) and converted to first-strand cDNA using a HiScript II 1st Strand cDNA Synthesis Kit (Vazyme, Cat# R212-01). qPCR was performed with gene-specific primers and 2×RealStar Fast SYBR qPCR Mix (GenStar, Cat# A301-01) and analyzed on an ABI QuantStudio 5 instrument. *OsUBQ* (LOC_Os03g13170) for rice and the *MoActin* gene (MGG_03982) for *M. oryzae* were used as reference genes. Primers used for qPCR can be found in Supplementary Data 5.

## Proximity-labeling spectrometry with TurboID

Ten-day-old transgenic rice seedlings producing TurboID-tagged proteins were used to perform proximity labeling. The seedlings were starved via incubation for 24 h in distilled water, submerged in 50 μM biotin, and incubated for 12 h at room temperature. Seedlings were washed three times with distilled water and subjected to protein extraction with native protein buffer as used in the Co-IP assay. Crude protein samples were passed through a 0.45-μm filter and desalted using an AKTA purification system (GE Healthcare) as previously described[23]. Approximately 4 mL desalted protein solution was incubated with 50 μL streptavidin-coated magnetic beads (Thermo Fisher, Cat# 65001) overnight at 4 °C for affinity purification. The beads were washed four times with PBS. Proteins were eluted by boiling the beads for 10 min in elution buffer (50 mM Tris-HCl pH 7.5, 150 mM NaCl, 1% [w/v] SDS, and 50 μM biotin), separated by SDS-PAGE, and subjected to four-dimensional (4D)-label-free LC-MS/MS analysis[72] by Allwegene (Beijing). Ratiometric analysis of the MS data was performed using protein enrichment area (LFQ) as previously described[23]. The proteins detected are listed in Supplementary Data 4. Gene Ontology analysis and prediction of subcellular localization of candidate proteins were performed with CARMO and CELLO, respectively.

## LC-MS/MS

For LC-MS/MS analysis, protein samples were digested in-gel by trypsin using the filter-aided proteome preparation (FASP) method[73]. The resulting peptides were desalted using a C18 Cartridge, lyophilized, dissolved in 40 μL 0.1% (v/v) formic acid, and quantified based on absorption at 280 nm. The tryptic peptides were then separated on a C18 reverse phase column (Thermo scientific EASY column, Cat# ES902) at a flow rate 300 nL/min with a NanoElute (Bruker) HPLC system. The separated samples were analyzed using a timsTOF Pro (Bruker) spectrometer. The detection mode was set to positive ion mode, and the ion source voltage was set to 1.5 kV. MS and MS/MS spectra were collected and analyzed using TOF, with a scanning range set to 100−1700 *m/z*. The parallel cumulative serial fragmentation (PASEF) mode was used for data acquisition with ion mobility (1/K0) 0.6–1.6 Vs/cm², and a primary mass spectrum corresponding to 10 secondary spectra in PASEF mode was collected. The dynamic exclusion time of MS/MS scanning was set to 24 s to avoid repeated scanning of parent ions. The original MS/MS data were searched against the Rice Genome Annotation Project database (http://rice.uga.edu/) using MaxQuant (v 1.6.14) with the following criteria: tryptic specificity was required, two missed cleavages were allowed, MS/MS tolerance of 20 ppm, the fixed modification was set as carbamidomethyl (C), the variable modification was set as oxidation (M), the

fragment ion mass tolerance was set to 0.02 Da, and a false discovery rate <0.01 was used to confidently select peptides and proteins.

## Quantification and statistical analysis

All quantitative data in this study were analyzed using GraphPad Prism 7.0 and are shown as means ± standard deviation (s.d.), means ± standard error of the mean (s.e.m.), or boxplots showing the maxima, minima, interquartile range, and median. Significance tests were performed in GraphPad Prism 7.0 using two-tailed Student's $t$ test, one-way ANOVA or two-way ANOVA, where appropriate. All experiments were performed two to four times with similar results. Details of statistical analyses including analytical methods, $n$ values, and significance levels ($p$-values) can be found in the figures or figure legends.

## Reporting summary

Further information on research design is available in the Nature Portfolio Reporting Summary linked to this article.

## Data availability

The MS proteomics data have been deposited at the ProteomeXchange Consortium (http://proteomecentral.proteomexchange.org) via the iProX partner repository[74,75] with the dataset identifier PXD040074. All data supporting the findings of this work are available in the paper, Supplementary Information files, and repository platform. Source data are provided with this paper.

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

## Acknowledgements

We would like to thank Dr. Yangnan Gu (University of California, Berkeley) for critical reading of the manuscript and providing valuable suggestions, Dr. Jie Zhang (Institute of Microbiology, Chinese Academy of Sciences) and Dr. Suomeng Dong (Nanjing Agricultural University) for helpful discussions, and Dr. Dongping Lu (Shanghai Jiao Tong University) for sharing in vivo ubiquitination constructs. This work was financially supported by grants from the National Natural Science Foundation of China (31422045) and the Agricultural Science and Technology Innovation Program (ASTIP).

## Author contributions

W.L., G.L.W., Q.Y., Y.X., and X.S. designed the research; X.S., X.Xie., Y.G., J.Z., Z.G., K.Z., X.Xia., J.M., H.X., and N.N. performed the experiments; W.L. and X.S. wrote the manuscript with contributions from all authors.

## Competing interests

The authors declare no competing interests.
