## [Peer Review File · Nature Communications]

A fungal core effector exploits the OsPUX8B.2–OsCDC48-6 module to suppress plant immunityReviewer #1 (Remarks to the Author):

The manuscript by Shi et al., provides characterization of the effector MoNLE1 and associated host proteins. The authors clearly show MoNLE1 is required for *M. oryzae* infection and can suppress host plant defense. They also identify a PUX protein, OsPUX8B.2, that interacts with MoNLE1 and that when overexpressed leads to increased resistant to *M. oryzae*, which is opposite of MoNLE1 effector it interacts with. Interestingly, the splice variant OsPUX8B.1 is actually more resistant to *M. oryzae* the mechanism behind this is not resolved but I believe that is beyond the scope of this manuscript. They provide nice characterization of OsPUX8B.2 and robust identification of interactors including OsBHT using proximity labeling. Finally, they show that OsBHT is a substrate of OsPUX8B.2 and this is disrupted by MoNLE1. Overall, this is a well carried out and complete story on an very important plant disease that will be of broad interested to plant biologists and pathologists.

Here are a few mostly minor comments for the authors to consider.

- 1) Lines 177-183 seem preliminary and overinterpreted. I think that they have not ruled out that it is not also an autophagy adapter, and CB-5083 experiment shows small differences with no statistics. I do not think additional experiments are necessary but perhaps this paragraph can be reworded in the context of the data already generated.
- 2) Fig 4C. the level of OsPUX8B.2 after 3-MA or ConA shown in the blot is not consistent with that shown in the bar chart.
- 3) The methods section describing the proximity labeling experiment are lacking detail. How were the eluted proteins digested? What is 4D-label free LC-MS/MS? What HPLC and MS were used? How long was the gradient? What were the MS acquisition settings? What software and settings were used to search the spectra and quantify protein levels?
- 4) A legend tab describing the data in Extended Data Table 4 would be helpful. Right now it is hard to tell what the data in each column is. Is the p-val or p.adj columns what is used to assess significance? In particular, what are the data columns ending in "_centered". Does this mean normalized? If so are the "_ratio" columns not normalized? It would also be helpful to have columns listing the individual replicate values.
- 5) A fold-change 2 cutoff is used in addition to p-val. Fold-change is an inherently arbitrary cutoff that is not based in biology and leads to false negatives. Is there a justification for why 2-fold is used? Especially since the core set of common interactors from multiple PL experiments is used. Likely its best to use the overlap of significantly enriched in each experiment, without considering Fold Change.
- 6) Data Availability. Perhaps a supplemental table or an addition to the ProteomeXchange submission with a key describing the raw files and other information required to reanalyze the data would be beneficial.
- 7) Line 117 states "closest homologes" however genes are either homologous or not.
- 8) Line 121/122 states "dicot" but I believe eudicot is the intended meaning in a phylogenetic sense.
- 9) Fig 3 i,j: define in legend that MoNLE1-mCherry = E1-mCherry (or at least that is what I assume.
- 10) Line 248. Maybe should cite the first reports optimizing TurboID for plants as well: <https://www.nature.com/articles/s41467-019-11202-z> and <https://elifesciences.org/articles/47864>
- 11) While the manuscript is well written overall it will benefit from additional editing.

Reviewer #2 (Remarks to the Author):

This is an interesting study. The authors discussed the role of CDC48s, specifically focusing on the involvement of plant UBX-containing (PUX) proteins in immune responses. The effector MoNLE1 from the fungal pathogen *Magnaporthe oryzae* is identified as a core virulence factor that suppresses rice immunity by targeting the UBX domain of OsPUX8B.2. This interference leads to the destabilization of OsPUX8B.2, a protein that, along with OsCDC48-6, positively regulates plant immunity against blast fungus. Additionally, the cytoplasmic substrate OsBHT negatively impacts defense against blast infection. MoNLE1 is found to promote the nuclear migration and degradation

of OsPUX8B.2, disrupting its association with OsBHT. However, I would like the following questions to be addressed.

The expression of MoNLE1 is dramatically dropped at 42 hpi according to Fig. 1c. What is the consequence if the expression MoNLE1 stays at high level?

Is nuclear localisation required for MoNLE1 to function?

MoNLE1-GFP is observed in the plant nucleus. Is MoNLE1-GFP restricted in the nucleus of infected cell only, or it can move ahead to uninfected cell and prepare the cells for successful infection?

Why a different isolate RB22 was used for infection assays with the transgenic lines while another isolate GUY11 was used for MoNLE1 function and localisation assays? Would the authors observe similar results using the same strain?

Fig 1h showing overexpression of MoNLE1 in rice compromised PAMP triggered ROS generation. What does the cumulative ROS generation of each leaf disc look like? Fig 1h cannot reflect the difference between biological replicates. I would like to include a more informative plot showing data of each tested leaf disc.

MoNLE1 homologs in wheat or grass blast strains still interacted with OsPUX8B.2 (Fig. 2h), and all MoNLE1 homologs also interacted with the UBX domain truncated splicing isoform of wheat TaPUX8 (Fig. 2i). Can the authors use AlphaFold2 predict the interacting interface between MoNLE1 homologs and PUX8B? Is there a conserved binding interface? Also, why the interaction with OsPUX8B.2 using QDO while the binding with TaPUX8 using TDO? Is the binding affinity with TaPUX8 much weaker?

One of the main conclusions of this study is MoNLE1 can promote the nuclear migration of OsPUX8B.2, but the only evidence is the transient expression in protoplast (Fig 2g). Each protoplast containing both plasmids might have variation of plasmid copies. Is there any additional evidence to support this conclusion? The authors have obtained transgenic line MoNLE1-HA. Does OsPUX8B.2 mainly localise in the nucleus of this line?

A model proposed by the author (Extended Data Fig. 6) shows MoNLE1 bind to OsPUX8B.2 in the host cytoplasm before entering the nucleus. Is there any evidence for this? Note, the authors also found the subcellular localisation of OsPUX8B.1 and OsPUX8B.3 was also affected by MoNLE1 but there's no interaction between MoNLE1 and OsPUX8B.1.

Extended Data Fig. 4 c and d were swapped in the figure legend?

Extended Data Fig. 4 h, OsCDC48-6-HA-OE line #6 and #15 have similar expression level where the expression level in #15 is even higher, but while there is less protein detected in the western blot image?

Figure legend of Extended Data Fig. 4; "24 hpt" should be "24 hpi"?

Reviewer #3 (Remarks to the Author):

The present manuscript by Shi et al. identifies a core effector, MoNLE1, from *Magnaporthe oryzae* that dampens host immunity by targeting PUX8B.2 from rice. The authors identify that OsPUX8B.2 interacts with CDC48-6 from rice and constitutes a proteasome substrate. Both OsPUX8B.2 and OsCDC48-6 are positive regulators of plant immunity and thus MoNLE1 promotes the proteasome-mediated degradation of OsPUX8B.2. Using a proximity labelling approach the authors identify a putative substrate, BHT, of the OsPUX8B.2 – CDC48-6 module, which acts as a negative regulator of plant defence reactions. By promoting the degradation of PUB8B.2, MoNLE1 disturbs the association with OsBHT, which in turn is beneficial for infection.

Overall, this study gives new insights how UBX domain containing proteins and CDC48 regulate

plant immune reactions and how microbial effectors hijack this module to cause disease. The findings of the authors are interesting and very timely, as the ubiquitin-proteasome system has been shown to play a major role in plant immunity. Nevertheless, I have some specific comments that need to be addressed.

Introduction/Main:

It would be great to have a bit more background information on proteasome-mediated degradation and plant immunity, as well as how effectors target by citing relevant literature. Given also the broad role of CDC48 and PUX proteins it is also essential to give background information about their so far identified functions and substrates.

Figure 2:

- An additional experiment to confirm nuclear recruitment of PUX8B2 by NLE1 is required (e.g., nuclear fractionation upon effector expression of PUX8). Are the experiments performed in the presence of proteasome inhibitor BTZ? Maybe the cytosolic part is degraded faster.
- 2f input GFP blot is missing

Figure 3:

- The authors should make sure that all Y2H constructs are expressed
- 3d: deltaUBX is only weakly expressed (input) – should be the same level for all samples.
- 3f: I can see a faint band with deltaUBX

Given this, it is important to include other in planta interaction data, e.g., BIFC or split Luciferase to confirm interactions

- 3i: this experiment needs an additional in planta confirmation – BiFC+quantification or split LUC assays + quantification

Statistics for Ext 4b are missing.

How do CDC48 protein levels behave? CDC48 antibody is available (phytoab) Did the authors test it in rice? Would be interesting to monitor changes of CDC48 in rice, if PUX8 is not the autophagy receptor of CDC48.

Figure 4:

- Please provide for a) and b) higher exposure to show that proteins (B2 and variants) are actually expressed – or BTZ treatment? Gene expression analysis for a) and b) is required to confirm proper expression of constructs.
- Fig4d: AvrPi9-GFP or GFP labelling is not correct – the authors should repeat experiments with BTZ treatment or inhibiting CDC48 activity with CB
- 4f). "Interestingly, co-expression of MoNLE1 with OsPUX8B.2 in rice protoplasts followed by treatment with the translational inhibitor cycloheximide (CHX) led to decreased OsPUX8B.2 levels." Where is the CHX treatment in the figure?

Figure 6

- There is a labeling mistake (6e and d)
- 6g: Please include BTZ or CDC48 inhibition to verify proteasome/cdc48 dependent degradation of substrate

- 6h: another in planta interaction assay is required to confirm this result.
- Is BHT ubiquitinated? For instance, do the authors observe a higher molecular weight band with BTZ? If BHT is indeed a substrate it should be ubiquitinated. It is possible to verify ubiquitination using TUBEs or the Ubiquitin pan selector tool. It would add more confidence that BHT is a real substrate of the PUX-CDC48 module.

General comment: Are all previous IPs performed in the presence of a proteasome inhibitor, if NLE1 promotes degradation of PUX8?

Is NLE1 degraded by PUX-CDC48 module or how does it avoid degradation?

Why is ATG8 still interacting with PUX8B2, what is the role of this interaction?

TubroID should be performed in infection context to capture real targets.

Reviewer #4 (Remarks to the Author):

In their manuscript "A fungal core effector exploits the OsPUX8B.2–1 OsCDC48-6 module to suppress plant immunity", Shi and colleagues describe the function of the conserved Magnaporthe oryzae effector MoNLE1 in suppression of plant immunity. They show that fungal monle1 mutants have reduced virulence, while expression of MoNLE1 in rice plants leads to reduced PAMP-triggered immune responses and enhanced disease susceptibility. A yeast-2-hybrid screen revealed the rice protein PUX8B, and specifically only the isoform PUX8B.2, as an interactor of MoNLE1. OsPUX8.2B also interacts with OsCDC48-6, and MoNLE1 promotes this interaction. CDC48 homologs have been shown to facilitate protein degradation via the 26S proteasome pathway, and coexpression of CDC48-6 promotes PUX8B turnover, which can be blocked by the proteasome inhibitor MG132, suggesting that PUX8B.2 is also regulated by the proteasome. Further, OsBHT was identified as an interactor of PUX8B.2. Coexpression of OsPUX8B.2 and OsCDC48-6 decreases OsBHT stability, suggesting that OsBHT is also targeted by this complex for proteasomal degradation. Finally, the authors show that OsBHT has a role in immunity, as Osbht mutants show enhanced disease resistance while overexpression of OsBHT results in enhanced susceptibility.

The data presented in the manuscript led the authors to propose an interesting model of how a pathogen effector leverages CDC48 proteins and the plant 26S proteasome system to suppress immune signaling and promote virulence. According to the model, OsBHT protein is targeted for proteasome-mediated protein turnover by OsPUX8B.2 and OsCDC48-6, keeping protein levels low. During Magnaporthe oryzae infection, the effector MoNLE1 disrupts the interaction between OsBHT and OsPUX8B.2, allowing OsBHT protein to accumulate and negatively regulate immunity. Given the conservation of NLE1 in M. oryzae isolates from diverse hosts, and the fact that these homologs interact with PUX8B.2 isoforms from both rice and wheat, this mechanism may be of general importance for M. oryzae infection, which would be of interest to researchers in the plant immunity field. The methods employed are appropriate, and the experiments appear to be technically sound.

However, I believe the authors' proposed model is not yet sufficiently supported by the data presented, and additional experiments, outlined below, are required to reinforce their conclusions.

1. The relationship between OsPUX8B.2 and OsBHT is still unclear. While the authors propose that a OsPUX8B.2-OsCDC48-6 complex targets OsBHT for degradation, the data does not rule out that OsBHT may promote OsPUX8B.2 degradation. The presence of SGT1-like domains in OsBHT would actually be consistent with a role in regulating protein stability. In Fig. 6g, the authors show that co-expression with OsPUX8B.2 alone does not substantially reduce OsBHT stability, and that OsCDC48-6 is required. However, they should also test OsCDC48-6 expression alone to determine whether OsPUX8B.2 contributes to the increased OsBHT stability or whether OsCDC48-6 alone is sufficient.

2. Their model is contradicted somewhat by the finding that the *Ospux8b* mutant does not show altered resistance. According to the model, *OsBHT* should accumulate more in *Ospux8b* plants and result in enhanced disease susceptibility. The authors should examine the *OsBHT* protein level in the *Ospux8b*.

3. Plants overexpressing *OsPUX8B.1* show enhanced disease susceptibility, in contrast to the enhanced resistance of *OsPUX8B.2* overexpression. The authors speculate that the two isoforms may compete for the same substrates. However, they fail to find any interaction between *OsPUX8B.1* and *OsBHT*, so if *OsBHT* is an important substrate of *OsPUX8B.2*, it is still unclear how *OsPUX8B.1* regulates immunity. The stability of *OsBHT* protein should be examined in the *OsPUX8B.1* overexpression lines. Another explanation is that *OsPUX8B.1* overexpression affects *OsPUX8B.2* expression through a feedback mechanism, so the expression of *OsPUX8B.2* should also be examined in these plants.

4. When showing differential protein abundance, such as Fig. 4f, the transcript levels should also be shown, the rule out differences in regulation at level of transcription or mRNA stability.

5. To further confirm that *OsPUX8B.2* protein turnover is mediated by the 26S proteasome, the authors could look for polyubiquitinated *OsPUX8B.2*, particularly after MG132 treatment.

6. The mechanism and significance of the MoNLE1-mediated nuclear migration of *OsPUX8B* proteins is not clear. It is surprising that the other *OsPUX8B* isoforms also show nuclear migration when coexpressed with MoNLE1, even though they were not found to interact with the effector. Expression of MoNLE1 without the NLS would help to determine whether the nuclear localization of the effector is required for the nuclear migration of *OsPUX8B* proteins. Similarly, expression of the *OsPUX8B* isoforms with NLS or nuclear export signals would show whether the localization of *OsPUX8B* is important for its effect on *OsBHT* stability and disease resistance.

Some minor points:

7. I do not find *OsCDC48-6* or *OsATG8* in the list of hits from the TurboID experiment. Can the authors speculate why these were not identified?

8. The figure legend for extended data Fig. 1 is missing the description for the synteny plot in panel c. I am not sure if the given description for panel c is also for panel d, or if that panel is missing.

9. On line 90, the authors refer to the "signaling peptide" of MoNLE1. I guess this should be "signal peptide".

Reviewer #1 (Remarks to the Author):

The manuscript by Shi et al., provides characterization of the effector MoNLE1 and associated host proteins. The authors clearly show MoNLE1 is required for *M. oryzae* infection and can suppress host plant defense. They also identify a PUX protein, OsPUX8B.2, that interacts with MoNLE1 and that when overexpressed leads to increased resistant to *M. oryzae*, which is opposite of MoNLE1 effector it interacts with. Interestingly, the splice variant OsPUX8B.1 is actually more resistant to *M. oryzae* the mechanism behind this is not resolved but I believe that is beyond the scope of this manuscript. They provide nice characterization of OsPUX8B.2 and robust identification of interactors including OsBHT using proximity labeling. Finally, they show that OsBHT is a substrate of OsPUX8B.2 and this is disrupted by MoNLE1. Overall, this is a well carried out and complete story on an very important plant disease that will be of broad interested to plant biologists and pathologists.

Thank you for the positive comments.

Here are a few mostly minor comments for the authors to consider.

1) Lines 177-183 seem preliminary and overinterpreted. I think that they have not ruled out that it is not also an autophagy adapter, and CB-5083 experiment shows small differences with no statistics. I do not think additional experiments are necessary but perhaps this paragraph can be reworded in the context of the data already generated.

Response: We agree with this point. Our data indeed do not exclude the possibility that, although the UBX domain is required for both interactions, OsPUX8B.2 can simultaneously bind to OsATG8 and OsCDC48-6. Therefore, we rephrased the statement as “The finding that the UBX domain is required for interactions with both OsATG8a and OsCDC48-6 raised the possibility that OsPUX8B.2 might function differently from Arabidopsis PUX8.” We appreciate this insightful suggestion.

2) Fig 4C. the level of OsPUX8B.2 after 3-MA or ConA shown in the blot is not consistent with that shown in the bar chart.

Response: We have repeated this experiment and performed an RT-PCR analysis to exclude the possibility that the transcript levels were affected by these inhibitors, as Reviewers 3 and 4 suggested. The bar chart shows the relative protein levels of 4×HA-OsPUX8B.2 using data from three biological replicates. Please see the revised Fig. 4c.

3) The methods section describing the proximity labeling experiment are lacking detail. How were the eluted proteins digested? What is 4D-label free LC-MS/MS? What HPLC and MS were used? How long was the gradient? What were the MS acquisition settings? What software and settings were used to search the spectra and quantify protein levels?

Response: The missing details were added to Methods under the section “LC-MS/MS” and are copied below for your reference:

“For LC-MS/MS analysis, protein samples were digested in-gel by trypsin using the filter-aided proteome preparation (FASP) method⁷³. The resulting peptides were desalted using a C18 Cartridge,

lyophilized, dissolved in 40 μ L 0.1% (v/v) formic acid, and quantified based on absorption at 280 nm. The tryptic peptides were then separated on a C18 reverse phase column (Thermo scientific EASY column, Cat# ES902) at a flow rate 300 nL/min with a NanoElute (Bruker) HPLC system. The separated samples were analyzed using a timsTOF Pro (Bruker) spectrometer. The detection mode was set to positive ion mode, and the ion source voltage was set to 1.5 kV. MS and MS/MS spectra were collected and analyzed using TOF, with a scanning range set to 100–1700 m/z . The parallel cumulative serial fragmentation (PASEF) mode was used for data acquisition with ion mobility ($1/K0$) 0.6–1.6 Vs/cm², and a primary mass spectrum corresponding to 10 secondary spectra in PASEF mode was collected. The dynamic exclusion time of MS/MS scanning was set to 24 s to avoid repeated scanning of parent ions. The original MS/MS data were searched against the Rice Genome Annotation Project database (<http://rice.uga.edu/>) using MaxQuant (v 1.6.14) with the following criteria: tryptic specificity was required, two missed cleavages were allowed, MS/MS tolerance of 20 ppm, the fixed modification was set as carbamidomethyl (C), the variable modification was set as oxidation (M), the fragment ion mass tolerance was set to 0.02 Da, and a false discovery rate < 0.01 was used to confidently select peptides and proteins.”

4) A legend tab describing the data in Extended Data Table 4 would be helpful. Right now it is hard to tell what the data in each column is. Is the p-val or p.adj columns what is used to assess significance? In particular, what are the data columns ending in “_centered”. Does this mean normalized? If so are the “_ratio” columns not normalized? It would also be helpful to have columns listing the individual replicate values.

Response: We are sorry for not having included this information in our earlier submission. We have added a legend to the table, which now indicates that “p.adj < 0.05” and “log₂(FC) > 1” were used as cutoffs for selection of putative preys. All analyses were performed in R using the DEP package. According to DEP, the centered columns contain the average log₂ fold changes of LFQ scaled by protein-wise centering. We have checked the code used for analysis, and we are sure that all the calculations including the “_ratio” (log₂foldchange) were based on normalized datasets. To avoid any confusion, we provided the normalized LFQ of individual replicate instead of the centered value. Please see the revised Supplementary Data 4.

5) A fold-change 2 cutoff is used in addition to p-val. Fold-change is an inherently arbitrary cutoff that is not based in biology and leads to false negatives. Is there a justification for why 2-fold is used? Especially since the core set of common interactors from multiple PL experiments is used. Likely its best to use the overlap of significantly enriched in each experiment, without considering Fold Change.

Response: Fold-change and adjusted *p*-value were used in our experiments for the “enrichment” of putative preys. The cutoff Log₂(FC) >1 is indeed arbitrary but was applied here with reference to previously published work (Huang et al., 2020; Tang et al., 2020). Although higher or lower values have been used in other studies (Jia et al., 2023; Zhang et al., 2019), we think that a fold-change of 2 is a moderate criterion that provides a good compromise between false positives and false negatives. Solely relying on the overlap between replicates without considering the fold-change values may not be applicable because proteins that are more abundant in the control samples could also be selected. In fact, the best way to determine the appropriate fold-change cutoffs is to use known interacting and non-interacting proteins as reference standards, as described by Alice Y.

Ting's group (Qin et al., 2023). However, as OsPUX8B.2 has not been reported elsewhere until this study, and because we failed to identify OsCDC48-6 or OsATG8 in the MS data, this method is not applicable here. We are currently working on optimizing the PL approach in rice and hope to define superior means to determine these cutoffs.

References:

Huang, A., Tang, Y., Shi, X., Jia, M., Zhu, J., Yan, X., Chen, H., and Gu, Y. (2020). Proximity labeling proteomics reveals critical regulators for inner nuclear membrane protein degradation in plants. *Nat Commun* *11*, 3284.

Jia, M., Chen, X., Shi, X., Fang, Y., and Gu, Y. (2023). Nuclear transport receptor KA120 regulates molecular condensation of MAC3 to coordinate plant immune activation. *Cell Host Microbe* *31*, 1685-1699 e1687.

Qin, W., Cheah, J.S., Xu, C., Messing, J., Freibaum, B.D., Boeynaems, S., Taylor, J.P., Udeshi, N.D., Carr, S.A., and Ting, A.Y. (2023). Dynamic mapping of proteome trafficking within and between living cells by TransitID. *Cell* *186*, 3307-3324 e3330.

Tang, Y., Huang, A.B., and Gu, Y.N. (2020). Global profiling of plant nuclear membrane proteome in *Arabidopsis*. *Nature Plants* *6*, 838-847.

Zhang, Y., Song, G., Lal, N.K., Nagalakshmi, U., Li, Y., Zheng, W., Huang, P.J., Branon, T.C., Ting, A.Y., Walley, J.W., *et al.* (2019). TurboID-based proximity labeling reveals that UBR7 is a regulator of NLR immune receptor-mediated immunity. *Nat Commun* *10*, 3252.

6) Data Availability. Perhaps a supplemental table or an addition to the ProteomeXchange submission with a key describing the raw files and other information required to reanalyze the data would be beneficial.

Response: We have added another sheet in Supplementary Data 4 to indicate the data availability. Thanks for your suggestion.

7) Line 117 states “closest homologes” however genes are either homologous or not.

Response: We have removed “closest.”

8) Line 121/122 states “dicot” but I believe eudicot is the intended meaning in a phylogenetic sense.

Response: We have changed “dicot” to “eudicot” in the revised manuscript and in Supplementary Fig. 3c. Thank you for pointing this out.

9) Fig 3 i,j: define in legend that MoNLE1-mCherry = E1-mCherry (or at least that is what I assume).

Response: We now spell out E1 to the full name of MoNLE1. This panel is now in Fig. 3k.

10) Line 248. Maybe should cite the first reports optimizing TurboID for plants as well: <https://www.nature.com/articles/s41467-019-11202-z> and <https://elifesciences.org/articles/47864>

Response: Citations of these two papers have now been added to the revised manuscript. Thank you.

11) While the manuscript is well written overall it will benefit from additional editing.

Response: We have further polished the text.

Reviewer #2 (Remarks to the Author):

This is an interesting study. The authors discussed the role of CDC48s, specifically focusing on the involvement of plant UBX-containing (PUX) proteins in immune responses. The effector MoNLE1 from the fungal pathogen *Magnaporthe oryzae* is identified as a core virulence factor that suppresses rice immunity by targeting the UBX domain of OsPUX8B.2. This interference leads to the destabilization of OsPUX8B.2, a protein that, along with OsCDC48-6, positively regulates plant immunity against blast fungus. Additionally, the cytoplasmic substrate OsBHT negatively impacts defense against blast infection. MoNLE1 is found to promote the nuclear migration and degradation of OsPUX8B.2, disrupting its association with OsBHT. However, I would like the following questions to be addressed.

Thanks for the positive comments.

1 The expression of MoNLE1 is dramatically dropped at 42 hpi according to Fig. 1c. What is the consequence if the expression MoNLE1 stays at high level?

Response: To mimic the constitutive expression of MoNLE1, we generated the *MoNLE1-GFP*-overexpressing blast strain (GUY11 background) driven by the *RIBOSOMAL PROTEIN 27 promoter (RP27p)* (Response Fig. 1a) and tested its pathogenicity on rice. Over-expression of MoNLE1 increased the virulence of the blast fungus (Response Fig. 1b), further supporting that MoNLE1 is an important core virulence effector. This data was added as Supplementary Fig.1f, g.

Response Fig. 1. Overexpressing MoNLE1-GFP increases pathogenicity of blast fungus. a, Validation of *RP27p:MoNLE1-GFP* transformants using immunoblot. Ponceau S was used as loading control. b, Pathogenicity test of *RP27p:MoNLE1-GFP* transformants on rice (cultivar CO39). Data were recorded at 7 d post inoculation (dpi) and are shown as boxplots displaying the maximum and minimum, first and third quantiles, and the median. n = number of independent biological samples. Statistical analysis was performed with one-way ANOVA followed by Tukey's test. * $p < 0.05$ and ** $p < 0.01$. Scale bar, 1 cm.

2 Is nuclear localisation required for MoNLE1 to function?

Response: To answer this question, we performed three experiments:

- 1) We examined the interaction between MoNLE1 NLS mutant proteins and OsPUX8B.2.
- 2) We co-expressed MoNLE1^{NES} (a MoNLE1 variant carrying an NES in place of the NLS) with OsPUX8B.2 to test whether this variant can affect the nuclear migration and degradation of OsPUX8B.2.
- 3) We evaluated whether heterologous expression of *MoNLE1^{NES}-HA (Ubp:MoNLE1^{NES}-HA)*

in rice affects plant immunity in the same manner as intact MoNLE1.

Although the NES variant of MoNLE1 still interacted with OsPUX8B.2 (Fig. 2d), it largely decreased the nuclear movement or degradation of OsPUX8B.2 (Fig. 2h). Importantly, unlike *Ubip:MoNLE1-HA* plants, the *Ubip:MoNLE1^{NES}-HA* plants did not show increased susceptibility to infection (Supplementary Fig. 2a-c). These experiments clearly indicate that the nuclear localization of MoNLE1 is essential for function.

3 MoNLE1-GFP is observed in the plant nucleus. Is MoNLE1-GFP restricted in the nucleus of infected cell only, or it can move ahead to uninfected cell and prepare the cells for successful infection?

Response: This is an interesting point. We only detected MoNLE1-GFP accumulation in the nucleus of infected cells. Response Fig. 2 below shows that MoNLE1-GFP fluorescence (showing the BICs and host nuclei) is observed only in infected cells and not in adjacent uninfected cells at 42 hours post inoculation (hpi). However, we cannot exclude the possibility that MoNLE1-GFP could still accumulate in adjacent cells, but at a level below the detection limit of the microscope. As *MoNLE1* was mainly expressed at 18–24 hpi, we speculate that MoNLE1 mainly functions at the early infection stage of *M. oryzae*.

Response Fig. 2. Localization of MoNLE1-GFP in inoculated rice tissues. Arrowheads indicate BICs; the asterisk indicates the nucleus of a rice cell.

4 Why a different isolate RB22 was used for infection assays with the transgenic lines while another isolate GUY11 was used for MoNLE1 function and localisation assays? Would the authors observe similar results using the same strain?

Response: GUY11 was used as a model strain to study blast genes in various previously published studies (Giraldo et al., 2013; Li et al., 2023; Ryder et al., 2023). Therefore, we initially used this strain to study the function of MoNLE1 in blast fungus. The rice cultivar Nipponbare (NPB) has a high-quality genome sequence available and is used for generating transgenic plants in most rice-related studies. However, in our experiments, isolate GUY11 showed weak virulence toward NPB (the background of all transgenic plants), while RB22 was much more virulent. Upon punch inoculation with GUY11, lesions on inoculated leaves barely expanded even over a long period of infection (15 dpi). Even so, similar to RB22, we still observed a virulence effect of MoNLE1 in *MoNLE1-HA* heterologous expressing (*Ubip:MoNLE1-HA*) plants after inoculation with GUY11, based on relative fungal biomass accumulation, as shown in Response Fig. 3, indicating that MoNLE1 functions similarly with different blast strains. We have added this result to Supplementary Fig. 1k in the revised manuscript. As all *MoNLE1*-knockout/complementation/overexpression strains were generated in the GUY11 background, we

tested the function or localization of MoNLE1 in the highly susceptible rice cultivar CO39 and examined phenotypes of transgenic plants using RB22.

Response Fig. 3. Rice plants expressing *MoNLE1-HA* are more susceptible to fungal infection. Data were recorded at 15 dpi and quantified by relative fungal biomass. Data were shown as boxplots displaying the maximum and minimum, first and third quartiles, and the median statistical analysis was performed with one-way ANOVA followed by Tukey's test. * $p < 0.05$ and ** $p < 0.01$.

References:

- Giraldo, M.C., Dagdas, Y.F., Gupta, Y.K., Mentlak, T.A., Yi, M., Martinez-Rocha, A.L., Saitoh, H., Terauchi, R., Talbot, N.J., and Valent, B. (2013). Two distinct secretion systems facilitate tissue invasion by the rice blast fungus *Magnaporthe oryzae*. *Nat Commun* 4, 1996.
- Li, G., Gong, Z., Dulal, N., Marroquin-Guzman, M., Rocha, R.O., Richter, M., and Wilson, R.A. (2023). A protein kinase coordinates cycles of autophagy and glutaminolysis in invasive hyphae of the fungus *Magnaporthe oryzae* within rice cells. *Nat Commun* 14, 4146.
- Ryder, L.S., Lopez, S.G., Michels, L., Eseola, A.B., Sprakel, J., Ma, W., and Talbot, N.J. (2023). A molecular mechanosensor for real-time visualization of appressorium membrane tension in *Magnaporthe oryzae*. *Nature microbiology* 8, 1508-1519.

5 Fig 1h showing overexpression of MoNLE1 in rice compromised PAMP triggered ROS generation. What does the cumulative ROS generation of each leaf disc look like? Fig 1h cannot reflect the difference between biological replicates. I would like to include a more informative plot showing data of each tested leaf disc.

Response: ROS accumulation is a dynamic process during PAMP treatments, with the main differences between genotypes usually occurring between 6 and 20 min. We calculated the total luminescence (representing the cumulative ROS accumulation) of each leaf disc. The statistical analysis revealed that, although the chitin treatment did not lead to a statistically significant difference relative to the control, *Ubip:MoNLE1-HA* plants showed lower average ROS accumulation upon PAMPs (flg22 and chitin) treatment than the wild type (Response Fig. 4 a and b). This data has been added in the revised Fig. 1h.

Response Fig. 4. Cumulative ROS production of *Ubip:MoNLE1-HA* plants upon PAMPs treatment.

6 MoNLE1 homologs in wheat or grass blast strains still interacted with OsPUX8B.2 (Fig. 2h), and all MoNLE1 homologs also interacted with the UBX domain truncated splicing isoform of wheat TaPUX8 (Fig. 2i). Can the authors use AlphaFold2 predict the interacting interface between MoNLE1 homologs and PUX8B? Is there a conserved binding interface? Also, why the interaction with OsPUX8B.2 using QDO while the binding with TaPUX8 using TDO? Is the binding affinity with TaPUX8 much weaker?

Response: To investigate the possible structures of MoNLE1 and PUX8 in complex, we predicted the MoNLE1–OsPUX8B.2 and MoNLE1–TaPUX8.2 structures using AlphaFold-Multimer (Evans et al., 2021). However, the prediction results returned a very low confidence score (pLDDT < 50, ipTM < 0.3), largely because MoNLE1 has no homologs in the reference database (Response Fig. 5a–d). Despite this low confidence, the predicted structure (top prediction) indicated that MoNLE1 mainly binds to C-terminal amino acids of the UBX domain through its N terminus (Response Fig. 5e), which is consistent with our results, but did not indicate any differences in binding affinity between the two complexes. Indeed, yeast two-hybrid assays showed that the interaction between MoNLE1 homologs from *M. oryzae* strains collected from other plant hosts and TaPUX8.2 is only strong enough to support growth on triple dropout (TDO) plates but not on quadruple dropout (QDO) plates, indicating that the binding between MoNLE1 and TaPUX8.2 may be weaker than that of MoNLE1 and OsPUX8B.2. We have added the QDO results in the revised figure and briefly introduced these data in the manuscript.

Response Fig. 5. Structure prediction of the MoNLE1–OsPUX8B.2 and MoNLE1–TaPUX8.2 complexes using AlphaFold-Multimer.

a, Predicted local distance difference test (pLDDT) scores of the top five MoNLE1–OsPUX8B.2 structures.

b, Predicted aligned error (PAE) maps of the top five MoNLE1–OsPUX8B.2 structures. A, MoNLE1; B, OsPUX8B.2.

c, pLDDT scores of the top five MoNLE1–TaPUX8.2 structures.

d, PAE maps of the top five MoNLE1–TaPUX8.2 structures. A, MoNLE1; B, TaPUX8.2.

e, Predicted structures of OsPUX8B.2–MoNLE1 and TaPUX8.2–MoNLE1.

Reference:

Evans, R., O'Neill, M., Pritzel, A., Antropova, N., Senior, A., Green, T., Žídek, A., Bates, R., Blackwell, S., Yim, J., et al. (2021). Protein complex prediction with AlphaFold-Multimer. bioRxiv

2021.10.04.463034.

7 One of the main conclusions of this study is MoNLE1 can promote the nuclear migration of OsPUX8B.2, but the only evidence is the transient expression in protoplast (Fig 2g). Each protoplast containing both plasmids might have variation of plasmid copies. Is there any additional evidence to support this conclusion? The authors have obtained transgenic line MoNLE1-HA. Does OsPUX8B.2 mainly localise in the nucleus of this line?

Response: We performed nuclear fractionation of protoplasts co-transfected with *Ubip:4×HA-OsPUX8B.2* and *35S:MoNLE1-GFP* or *35S:MoNLE1^{NES}-GFP*. Although all protoplasts were pretreated with MG132 10 h before performing nuclear fractionation, MoNLE1 still caused obvious degradation of OsPUX8B.2. As MoNLE1 can promote the degradation of OsPUX8B.2, we calculated the relative nuclear abundance of 4×HA-OsPUX8B.2 in the presence of free GFP, MoNLE1-GFP, or MoNLE1^{NES}-GFP. We determined that MoNLE1-GFP indeed promoted the nuclear accumulation of OsPUX8B.2 compared with GFP, while MoNLE1^{NES}-GFP was associated with decreased OsPUX8B.2 levels in nuclei (Response Fig. 6). This experiment clearly indicates that MoNLE1 promotes the nuclear migration of OsPUX8B.2, which is dependent on the NLS of MoNLE1. This finding was added to Fig. 2h in the revised manuscript.

Response Fig. 6. Nuclear fractionation of transfected protoplasts. The relative nuclear protein abundance of 4×HA-OsPUX8B.2 was calculated by comparing the relative nuclear (N) protein level (4×HA-OsPUX8B.2^N normalized to histone H3) with the relative total (T) protein level (4×HA-OsPUX8B.2^T normalized to β-Tubulin) using Image J software.

Thank you for the suggestion to observe OsPUX8B.2 localization in the *MoNLE1*-OE plants. We transfected protoplasts prepared from WT, *Ubip:MoNLE1-HA*, and *Ubip:MoNLE1^{NES}-HA* plants with the *Ubip:OsPUX8B.2-GFP* plasmid followed by treatment with MG132. However, we did not observe clear differences in OsPUX8B.2 localization among these cells (please see Response Fig. 7 below). We speculate that MoNLE1 does not accumulate to high enough levels in these OE plants to visibly affect OsPUX8B.2 movement and abundance.

Response Fig. 7. Subcellular localization of OsPUB8B.2-GFP in protoplasts of indicated plants.

8 A model proposed by the author (Extended Data Fig. 6) shows MoNLE1 bind to OsPUB8B.2 in the host cytoplasm before entering the nucleus. Is there any evidence for this? Note, the authors also found the subcellular localisation of OsPUB8B.1 and OsPUB8B.3 was also affected by MoNLE1 but there's no interaction between MoNLE1 and OsPUB8B.1.

Response: It would be difficult to test for an interaction between MoNLE1 and OsPUB8B.2 in the cytoplasm, as the nuclear translocation of MoNLE1 is fast. Indeed, we detected very low levels of MoNLE1 in the cytosolic fraction when performing nuclei/cytosol fractionation assays. However, since MoNLE1 promotes the nuclear migration of OsPUB8B.2, which is dependent on the NLS, and the MoNLE1^{NES} variant decreases OsPUB8B.2 degradation (please see our response to Point 7 and Response Fig. 6), we speculate that MoNLE1 can directly carry cytosolic OsPUB8B.2 to the nucleus for degradation. Accordingly, we have revised the model based on this finding (Fig. 7). Given the high similarity of the three OsPUB8 isoforms, we hypothesize that OsPUB8B.1 and OsPUB8B.3 may have common interactors that also associate with MoNLE1, which will be investigated in our future research but is outside the scope of this work. Our ongoing work has already identified additional interactors (such as rice 14-3-3 proteins) between MoNLE1 and the OsPUB8 proteins. We hope these new findings will help us to unravel these details in a future study.

9 Extended Data Fig. 4 c and d were swapped in the figure legend?

Response: These have been changed and are now shown in Supplementary Fig. 6. Thank you.

10 Extended Data Fig. 4 h, OsCDC48-6-HA-OE line #6 and #15 have similar expression level where the expression level in #15 is even higher, but while there is less protein detected in the western blot image?

Response: This difference may be due to the inability to obtain sufficient samples in the T₀ generation for proper quantification of protein levels. We re-examined the protein abundance of OsCDC48-6-HA in lines #6 and #15 using T₁ plants, which showed protein levels consistent with transcript levels. The result has been updated in the revised manuscript (now in Supplementary Fig. 6i).

11 Figure legend of Extended Data Fig. 4; “24 hpt” should be “24 hpi”?

Response: Here, hpt means hours post transfection. Thank you for your careful reading of this manuscript.

Reviewer #3 (Remarks to the Author):

The present manuscript by Shi et al. identifies a core effector, MoNLE1, from *Magnaporthe oryzae* that dampens host immunity by targeting PUX8B.2 from rice. The authors identify that OsPUX8B.2 interacts with CDC48-6 from rice and constitutes a proteasome substrate. Both OsPUX8B.2 and OsCDC48-6 are positive regulators of plant immunity and thus MoNLE1 promotes the proteasome-mediated degradation of OsPUX8B.2. Using a proximity labelling approach the authors identify a putative substrate, BHT, of the OsPUX8B.2 – CDC48-6 module, which acts as a negative regulator of plant defence reactions. By promoting the degradation of PUB8B.2, MoNLE1 disturbs the association with OsBHT, which in turn is beneficial for infection.

Overall, this study gives new insights how UBX domain containing proteins and CDC48 regulate plant immune reactions and how microbial effectors hijack this module to cause disease. The findings of the authors are interesting and very timely, as the ubiquitin-proteasome system has been shown to play a major role in plant immunity. Nevertheless, I have some specific comments that need to be addressed.

Thank you for the positive comments.

Introduction/Main:

1 It would be great to have a bit more background information on proteasome-mediated degradation and plant immunity, as well as how effectors target by citing relevant literature. Given also the broad role of CDC48 and PUX proteins it is also essential to give background information about their so far identified functions and substrates.

Response: We have rewritten the introduction and added more information. Please see lines 29–75 of the revised manuscript. Thank you for this suggestion.

2 Figure 2:

An additional experiment to confirm nuclear recruitment of PUX8B2 by NLE1 is required (e.g., nuclear fractionation upon effector expression of PUX8). Are the experiments performed in the presence of proteasome inhibitor BTZ? Maybe the cytosolic part is degraded faster.

Response: Please see Response Fig. 5 for the nuclear fractionation analysis in our Response to points 7 and 8 from Reviewer #2. We found that MoNLE1^{NES}-GFP was associated with decreased OsPUXB.2 levels in nuclei and also exhibited a weaker ability for promoting OPUX8B.2 turnover (Response Fig. 6). We therefore speculate that MoNLE1 mainly promotes OsPUX8B.2 degradation in the rice nucleus. The model was modified accordingly.

3 2f input GFP blot is missing

Response: The original GFP panel for the input has now been added in revised Fig. 2f.

4 Figure 3: The authors should make sure that all Y2H constructs are expressed

Response: We have checked that all proteins are produced and accumulate from the Y2H constructs in yeast; these data are shown in Supplementary Fig. 4.

5 Figure 3d: deltaUBX is only weakly expressed (input) – should be the same level for all samples.

6 Figure 3f: I can see a faint band with deltaUBX

Given this, it is important to include other in planta interaction data, e.g., BIFC or split Luciferase to confirm interactions

Response: We used a split luciferase complementation (SLC) assay to confirm these results, which has been added as Fig. 3e, h.

7 Figure 3i: this experiment needs an additional in planta confirmation – BiFC+quantification or split LUC assays + quantification

Response: We performed competitive SLC assays to confirm the Co-IP experiments. Specifically, we tested the interaction between OsPUX8B.2 and OsATG8a/OsCDC48-6 in the presence of GFP or MoNLE1-GFP, which showed that MoNLE1 did not significantly affect the association of OsPUX8B.2 with OsATG8a but increased the interaction of OsPUX8B.2 with OsCDC48-6, which is consistent with the Co-IP experiments. These data have been added as Fig. 3l in the revised manuscript.

8 Statistics for Ext 4b are missing.

Response: The statistical analysis has now been added in revised Supplementary Fig. 6b. Data were changed to bar plots to label significance levels.

9 How do CDC48 protein levels behave? CDC48 antibody is available (phytoab) Did the authors test it in rice? Would be interesting to monitor changes of CDC48 in rice, if PUX8 is not the autophagy receptor of CDC48.

Response: This antibody was generated with peptides derived from Arabidopsis CDC48A/D/E. We did not test the anti-CDC48 antibody in rice because it may recognize multiple CDC48 homologs. As our study mainly focused on OsCDC48-6, we opted against using this antibody. As an alternative, we monitored the protein level of OsCDC48-6 during *M. oryzae* infection (isolate RB22) using *OsCDC48-6-HA-OE* (#6) plants. The protein levels of OsCDC48-6-HA were not significantly affected by *M. oryzae* infection (Response Fig. 8), indicating that OsCDC48-6 is mainly regulated at the transcription level, as shown in Fig. 3m.

Response Fig. 8. Immunoblot analysis of OsCDC48-6-HA under control conditions and RB22 infection.

10 Figure 4:

Please provide for a) and b) higher exposure to show that proteins (B2 and variants) are actually expressed – or BTZ treatment? Gene expression analysis for a) and b) is required to confirm proper expression of constructs.

Response: In our experiments, OsPUXB.2 could not be detected by immunoblotting even when the construct was driven by the 35S promoter and even with high exposure. We tested different protein

degradation inhibitors before the first submission of our manuscript and found that MG132 resulted in the accumulation of some OsPUXB.2 (Response Fig. 9). However, because this result is not easily quantified, we instead used *Ubip:4×HA-OsPUX8B.2* to confirm this finding (shown in Fig 4c). For clarity, we did not show the original result (Response Fig. 9) in the manuscript. Nevertheless, we have repeated these two experiments using immunoblotting and RT-PCR analysis and confirmed that all corresponding genes are transcribed and that their encoded proteins accumulate (please see the new version of Fig. 4a, b).

Response Fig. 9. MG132 treatment results in the accumulation of OsPUX8B.2-HA.

11 Figure 4d: AvrPi9-GFP or GFP labelling is not correct – the authors should repeat experiments with BTZ treatment or inhibiting CDC48 activity with CB

Response: Sorry for the incorrect description in the legend of Fig. 4d. GFP but not AvrPi9-GFP was used as a control. We have revised it in the new version of manuscript. We have repeated this experiment with both MG132 and CB-5083. Moreover, we included RT-PCR analysis to rule out any effect of transcript levels. This result is shown in Fig. 4e.

12 Figure 4f). “Interestingly, co-expression of MoNLE1 with OsPUX8B.2 in rice protoplasts followed by treatment with the translational inhibitor cycloheximide (CHX) led to decreased OsPUX8B.2 levels.” Where is the CHX treatment in the figure?

Response: CHX was added to all protoplasts. We repeated this experiment using immunoblotting and RT-PCR analysis. Please see revised Fig. 4g.

13 Figure 6

There is a labeling mistake (6e and d)

Response: We have revised this figure. Thank you for your careful reading.

14 Figure 6g: Please include BTZ or CDC48 inhibition to verify proteasome/cdc48 dependent degradation of substrate

Response: We repeated this experiment in the presence of MG132 or CB-5083. We also tested whether OsATG8a or OsCDC48-6 alone is sufficient to affect OsBHT stability, as Reviewer 4 suggested. Our results indicate that OsBHT is only visibly destabilized in the presence of both OsPUX8B.2 and OsCDC48-6, while MG132 or CB-5083 treatment stabilizes OsBHT. RT-PCR analysis revealed no change in *OsBHT-cMyc* transcript levels. Please see revised Fig. 6e.

15 Figure 6h: another in planta interaction assay is required to confirm this result.

Response: We performed a competitive SLC assay and observed that MoNLE1 significantly blocks the association of OsPUX8B.2 with OsBHT. This result was added to the revised manuscript as Fig.

6g.

16 Is BHT ubiquitinated? For instance, do the authors observe a higher molecular weight band with BTZ? If BHT is indeed a substrate it should be ubiquitinated. It is possible to verify ubiquitination using TUBEs or the Ubiquitin pan selector tool. It would add more confidence that BHT is a real substrate of the PUX-CDC48 module.

Response: We have indeed observed lower mobility forms of OsBHT under higher exposure (Response Fig. 10a), indicating that OsBHT is modified. To test whether OsBHT is ubiquitinated, we performed an *in vivo* ubiquitination assay (Zhou et al., 2014; Liu et al., 2022) by co-expressing *OsBHT-HA* and *FLAG-UBQ* in rice protoplasts, followed by MG132 treatment. OsBHT-HA was co-immunoprecipitated by FLAG-UBQ (Response Fig. 10b), indicating that OsBHT is truly ubiquitinated. Lower mobility forms of OsBHT were not clearly observed in IP samples probably because of low levels of poly-ubiquitination. These data were also added to the revised manuscript as Supplementary Fig. 7h.

Response Fig. 10. OsBHT is ubiquitinated in rice protoplasts. a, Lower mobility forms of OsBHT were observed upon higher exposure. b, *In vivo* ubiquitination assay showing that OsBHT is ubiquitinated in rice protoplasts transfected with the indicated constructs.

References:

Zhou, J., He, P., and Shan, L. (2014). Ubiquitination of plant immune receptors. *Methods in molecular biology* (Clifton, NJ) 1209, 219-231.

Liu, X., Zhou, Y., Du, M., Liang, X., Fan, F., Huang, G., Zou, Y., Bai, J., and Lu, D. (2022). The calcium-dependent protein kinase CPK28 is targeted by the ubiquitin ligases ATL31 and ATL6 for proteasome-mediated degradation to fine-tune immune signaling in Arabidopsis. *Plant Cell* 34, 679-697.

17 General comment: Are all previous IPs performed in the presence of a proteasome inhibitor, if NLE1 promotes degradation of PUX8?

Response: Correct, all samples used for IP assays were pretreated with 50 μ M MG132 before protein extraction. This information has been added to the Methods.

18 Is NLE1 degraded by PUX-CDC48 module or how does it avoid degradation?

Response: This is an excellent question! We examined the status of MoNLE1 in the presence of

OsPUX8B.2 and OsCDC48-6 and surprisingly found that MoNLE1 can also be degraded by OsPUX8B.2 and OsCDC48-6; this degradation can be blocked by MG132 treatment and weakly inhibited by CB-5083 treatment. This experiment further strengthens the notion that the OsPUX8B.2–OsCDC48-6 module positively regulates defense through targeting the *M. oryzae* core effector MoNLE1 and that MoNLE1 and OsPUX8B.2 can mutually deplete each other during the rice–blast fungus interaction. These data support the conclusions of this study and have been added as Fig. 4h. We also revised our model according to these new findings (Fig. 7).

19 Why is ATG8 still interacting with PUX8B2, what is the role of this interaction?

Response: It was recently reported that *M. oryzae* infection can induce autophagy in rice cells (Liang et al., 2023), which is important for activation of plant immune responses. We believe that the interactions between OsPUX8B.2 and ATG8 homologs are involved in autophagy-related functions, with OsPUX8B.2 functioning as a selective adaptor for specific substrates other than OsBHT. It is possible that OsPUX8B.2 participates in both autophagy and CDC48–26S proteasome pathways. However, as MoNLE1 did not disturb the association between OsPUX8B.2 and OsATG8s (Fig. 3k, l), we focused on the relationships among MoNLE1, OsPUX8B.2, and OsCDC48-6 in this work.

Reference:

Liang, Y., Meng, F., Zhao, X., He, X., and Liu, J. (2023). OsHLP1 is an endoplasmic-reticulum-phagy receptor in rice plants. *Cell reports* 42, 113480.

20 TurboID should be performed in infection context to capture real targets.

Response: When performing proximity labeling, we need to submerge rice seedlings producing the corresponding proteins in 50 μ M biotin. Moreover, rice leaves are more hydrophobic than those of *Arabidopsis*, which can limit the labeling efficiency. Therefore, it is challenging to do this experiment during infection. We are working on optimizing this method for rice. We will try to add biotin to the spore suspension and test whether it can work during infection. Thank you for this suggestion.

Reviewer #4 (Remarks to the Author):

In their manuscript “A fungal core effector exploits the OsPUX8B.2–OsCDC48-6 module to suppress plant immunity”, Shi and colleagues describe the function of the conserved Magnaporthe *oryzae* effector MoNLE1 in suppression of plant immunity. They show that fungal *monle1* mutants have reduced virulence, while expression of MoNLE1 in rice plants leads to reduced PAMP-triggered immune responses and enhanced disease susceptibility. A yeast-2-hybrid screen revealed the rice protein PUX8B, and specifically only the isoform PUX8B.2, as an interactor of MoNLE1. OsPUX8.2B also interacts with OsCDC48-6, and MoNLE1 promotes this interaction. CDC48 homologs have been shown to facilitate protein degradation via the 26S proteasome pathway, and coexpression of CDC48-6 promotes PUX8B turnover, which can be blocked by the proteasome inhibitor MG132, suggesting that PUX8B.2 is also regulated by the proteasome. Further, OsBHT was identified as an interactor of PUX8B.2. Coexpression of OsPUX8B.2 and OsCDC48-6 decreases OsBHT stability, suggesting that OsBHT is also targeted by this complex for proteasomal

degradation. Finally, the authors show that OsBHT has a role in immunity, as *Os**b**ht* mutants show enhanced disease resistance while overexpression of OsBHT results in enhanced susceptibility.

The data presented in the manuscript led the authors to propose an interesting model of how a pathogen effector leverages CDC48 proteins and the plant 26S proteasome system to suppress immune signaling and promote virulence. According to the model, OsBHT protein is targeted for proteasome-mediated protein turnover by OsPUX8B.2 and OsCDC48-6, keeping protein levels low. During *Magnaporthe oryzae* infection, the effector MoNLE1 disrupts the interaction between OsBHT and OsPUX8B.2, allowing OsBHT protein to accumulate and negatively regulate immunity. Given the conservation of NLE1 in *M. oryzae* isolates from diverse hosts, and the fact that these homologs interact with PUX8B.2 isoforms from both rice and wheat, this mechanism may be of general importance for *M. oryzae* infection, which would be of interest to researchers in the plant immunity field. The methods employed are appropriate, and the experiments appear to be technically sound.

Thank you for the positive comments.

However, I believe the authors' proposed model is not yet sufficiently supported by the data presented, and additional experiments, outlined below, are required to reinforce their conclusions.

1. The relationship between OsPUX8B.2 and OsBHT is still unclear. While the authors propose that a OsPUX8B.2-OsCDC48-6 complex targets OsBHT for degradation, the data does not rule out that OsBHT may promote OsPUX8B.2 degradation. The presence of SGT1-like domains in OsBHT would actually be consistent with a role in regulating protein stability. In Fig. 6g, the authors show that co-expression with OsPUX8B.2 alone does not substantially reduce OsBHT stability, and that OsCDC48-6 is required. However, they should also test OsCDC48-6 expression alone to determine whether OsPUX8B.2 contributes to the increased OsBHT stability or whether OsCDC48-6 alone is sufficient.

Response: To assess whether OsBHT acts as a co-chaperone to affect OsPUXB.2 stability, we co-expressed *OsBHT-GFP* and *4×HA-OsPUX8B.2* in rice protoplasts without any treatment. This experiment showed that OsBHT only moderately stabilizes OsPUX8B.2 (Response Fig. 11). These data were added to the revised manuscript as Supplementary Fig. 7g.

Response Fig. 11. Moderate stabilization of OsBHT by OsPUX8B.2.

To further examine the degradation of OsBHT by OsPUX8B.2 and OsCDC48-6, we repeated the experiment with each protein individually and included treatment with degradation inhibitors and RT-PCR analysis. We determined that OsBHT-cMyc can only be degraded in the presence of OsPUX8B.2 and OsCDC48-6, and this degradation can be blocked by MG132 or CB-5083 treatment (Response Fig. 12). These data were added to the revised manuscript as Fig. 6e.

Response Fig. 12. Effect of OsPUX8B.2, OsCDC48-6, and proteasome inhibitor treatment on OsBHT stability.

2. Their model is contradicted somewhat by the finding that the *Ospux8b* mutant does not show altered resistance. According to the model, OsBHT should accumulate more in *Ospux8b* plants and result in enhanced disease susceptibility. The authors should examine the OsBHT protein level in the *Ospux8b*.

Response: The normal resistance of *Ospux8b* mutants can be explained by the following ideas.

1) The basal level of OsPUX8B.2 is kept at quite a low level, as evidenced by its high instability and low transcript level (Fig. 4a-c and Supplementary Fig. 5d) under normal conditions. This means that the protein level of OsPUX8B.2 may vary little in *Ospux8b* mutants compared with the wild type, resulting in no difference in OsBHT abundance in *Ospux8b* mutants. In agreement, we did not observe clear differences in OsBHT-cMyc levels between the *OsBHTpro:OsBHT-cMyc* and *Ospux8b-2 OsBHTpro:OsBHT-cMyc* transgenic lines (Response Fig. 13).

2) OsPUX8B.1 may function redundantly with its homolog OsPUX8A, with which it shares high sequence similarity (Supplementary Fig. 3a). The loss of OsPUX8B.1 function in the *Ospux8b* mutant may thus be compensated for by OsPUX8A.

Response Fig. 13. Abundance of OsBHT in the wild type and the *Ospux8b.2* mutant background. The arrowhead indicates the expected size of OsBHT-cMyc; rice HSP82 was used as a loading control.

3. Plants overexpressing OsPUX8B.1 show enhanced disease susceptibility, in contrast to the enhanced resistance of OsPUX8B.2 overexpression. The authors speculate that the two isoforms

may compete for the same substrates. However, they fail to find any interaction between OsPUX8B.1 and OsBHT, so if OsBHT is an important substrate of OsPUX8B.2, it is still unclear how OsPUX8B.1 regulates immunity. The stability of OsBHT protein should be examined in the OsPUX8B.1 overexpression lines. Another explanation is that OsPUX8B.1 overexpression affects OsPUX8B.2 expression through a feedback mechanism, so the expression of OsPUX8B.2 should also be examined in these plants.

Response: Good point! To determine whether OsBHT is regulated by OsPUXB.1, we transfected *OsBHT-GFP* into protoplasts prepared from wild-type, *OsPUX8B.1-HA-OE*, and *OsPUX8B.2-HA-OE* plants. With a CHX treatment (100 μ M) of 10 h before collection of protoplasts for protein extraction, we observed that OsBHT-GFP abundance is comparable between *OsPUX8B.1-HA-OE* plants and the wild type but is much lower in *OsPUX8B.2-HA-OE* plants, indicating that OsBHT abundance is regulated by OsPUX8B.2 but not by OsPUX8B.1 (Response Fig. 14a). These data have been added as Supplementary Fig. 7i in the revised manuscript. Moreover, the transcript levels of *OsPUX8B.2* were not altered in *OsPUX8B.1-HA-OE* plants (Response Fig. 14b). These data suggest that OsPUX8B.1 might not be able to regulate OsBHT either directly or indirectly through OsPUX8B.2 to participate in plant immunity, which means that the phenotype of *OsPUX8B.1-HA-OE* plants cannot be simply explained by OsBHT. This is one of the limitations of this study that we discuss in the revised manuscript. We hypothesize that OsPUX8B.2 but not OsPUX8B.1 may have other common interactors in addition to OsBHT (please also see Response to point 8 from Reviewer #2). This work mainly focused on the relationship between MoNLE1, OsPUX8B.2, and OsCDC48-6; however, we hope to unveil the regulatory mechanism between OsPUX8B.1 and OsPUX8B.2 in future studies.

Response Fig. 14. Differential effect of OsPUX8B.1 and OsPUX8B.2 on OsBHT abundance. a, OsBHT is more unstable in protoplasts prepared from *OsPUX8B.2-HA-OE* plants than in those from WT plants. b, Expression levels of *OsPUX8B.1* and *OsPUX8B.2* in the indicated plants.

4. When showing differential protein abundance, such as Fig. 4f, the transcript levels should also be shown, the rule out differences in regulation at level of transcription or mRNA stability.

Response: We appreciate this suggestion. It is necessary to rule out the possibility of changes at the transcript level. Therefore, we have repeated almost all of the experiments related to protein abundance and determined the mRNA levels of the indicated constructs. Please see revised Figs. 4 and 6.

5. To further confirm that OsPUX8B.2 protein turnover is mediated by the 26S proteasome, the authors could look for polyubiquitinated OsPUX8B.2, particularly after MG132 treatment.

Response: To answer this question, we performed an *in vivo* ubiquitination assay (Zhou et al., 2014; Liu et al., 2022) in rice protoplasts. We co-expressed 4×HA-OsPUX8B.2 and FLAG-UBQ in rice protoplasts, followed by treatment with MG132. After immunoprecipitation with FLAG beads, we indeed observed that 4×HA-OsPUX8B.2 was bound by ubiquitin (Response Fig. 15). The smear above the main band suggests that OsPUX8B.2 is polyubiquitinated. This result was added as Fig. 4d in the revised manuscript.

Response Fig. 15. Ubiquitination assay of OsPUX8B.2.

References:

Zhou, J., He, P., and Shan, L. (2014). Ubiquitination of plant immune receptors. *Methods in molecular biology* (Clifton, NJ) 1209, 219-231.

Liu, X., Zhou, Y., Du, M., Liang, X., Fan, F., Huang, G., Zou, Y., Bai, J., and Lu, D. (2022). The calcium-dependent protein kinase CPK28 is targeted by the ubiquitin ligases ATL31 and ATL6 for proteasome-mediated degradation to fine-tune immune signaling in Arabidopsis. *Plant Cell* 34, 679-69

6. The mechanism and significance of the MoNLE1-mediated nuclear migration of OsPUX8B proteins is not clear. It is surprising that the other OsPUX8B isoforms also show nuclear migration when coexpressed with MoNLE1, even though they were not found to interact with the effector. Expression of MoNLE1 without the NLS would help to determine whether the nuclear localization of the effector is required for the nuclear migration of OsPUX8B proteins. Similarly, expression of the OsPUX8B isoforms with NLS or nuclear export signals would show whether the localization of OsPUX8B is important for its effect on OsBHT stability and disease resistance.

Response: Please see Response Figure 6 for the nuclear fractionation analysis in our Response to point 7 from Reviewer #2.

Given the high similarity between the three OsPUX8B isoforms, we hypothesize that OsPUX8B.1 and OsPUX8B.3 may have common interactors in addition to MoNLE1, which will be investigated in our future research but is not within the scope of this work.

The OsPUX8B isoforms do not have typical NLSs and are present in both the nucleus and cytoplasm. Therefore, we were not able to generate transgenic plants that accumulate OsPUX8B variants that stay in the nucleus or cytoplasm. Given that OsBHT mainly localizes to the cytoplasm, which was

not affected by OsPUX8B.2, we speculate that interaction and degradation of OsBHT mainly take place in the cytoplasm.

Some minor points:

7. I do not find OsCDC48-6 or OsATG8 in the list of hits from the TurboID experiment. Can the authors speculate why these were not identified?

Response: We also noticed this. We propose three reasons to explain this situation.

1) The high instability of OsPUX8B.2 may only allow the labeling of highly abundant proteins.

2) OsCDC48-6 may be maintained at a low level and induced upon *M. oryzae* infection (Fig. 3m).

3) The low molecular weight of OsATG8 proteins (~14 kD) may mean that these proteins are lost during excision of protein bands from the gels for MS analysis. We are still working on optimizing the TurboID labeling approach for rice. We suggest that the labeling conditions (biotin concentration, labeling time, and temperature) should be pre-determined for different proteins.

8. The figure legend for extended data Fig. 1 is missing the description for the synteny plot in panel c. I am not sure if the given description for panel c is also for panel d, or if that panel is missing.

Response: We have revised the figure legend. Thank you.

9. On line 90, the authors refer to the “signaling peptide” of MoNLE1. I guess this should be “signal peptide”.

Response: Revised as suggested. Thank you.

Reviewer #1 (Remarks to the Author):

Thank you for addressing my comments. Nice work!

Reviewer #2 (Remarks to the Author):

Thank you for your efforts. The questions I raised for the first submission have been well addressed in the current version.

Reviewer #3 (Remarks to the Author):

The authors impressively addressed all of my concerns – congrats to this very nice work.

I have one minor point: I think that Supp. Figure 7h should move to the main Figure 6 as OsBHT is a substrate and this experiment strengthens the findings.

Reviewer #4 (Remarks to the Author):

The authors have thoroughly addressed the points that I raised in the first round of review.